# Contrasting Phylogeographic Patterns of Mitochondrial and Genome-Wide Variation in the Groundwater Amphipod *Crangonyx islandicus* That Survived the Ice Age in Iceland

David Eme [1,2,]*, Kristen M. Westfall [1,3], Brynja Matthíasardóttir [1,4], Bjarni Kristófer Kristjánsson [5] and Snæbjörn Pálsson [1,]*

1. Department of Life and Environmental Sciences, University of Iceland, Askja, Sturlugata 7, 101 Reykjavik, Iceland
2. RiverLy Research Unit, National Research Institute for Agriculture Food and Environment (INRAE), 5 rue de la Doua, 69100 Villeurbanne, France
3. Fisheries and Oceans Canada, Pacific Biological Station, Nanaimo, BC V9T 6N7, Canada
4. National Institutes of Health, NHGRI, 10 Center Drive, Bethesda, MD 20892, USA
5. Department of Aquaculture and Fish Biology, Hólar University, Háeyri 1, 550 Sauðárkrókur, Iceland
* Correspondence: david.eme@inrae.fr (D.E.); snaebj@hi.is (S.P.)

**Abstract:** The analysis of phylogeographic patterns has often been based on mitochondrial DNA variation, but recent analyses dealing with nuclear DNA have in some instances revealed mito-nuclear discordances and complex evolutionary histories. These enigmatic scenarios, which may involve stochastic lineage sorting, ancestral hybridization, past dispersal and secondary contacts, are increasingly scrutinized with a new generation of genomic tools such as RADseq, which also poses additional analytical challenges. Here, we revisited the previously inconclusive phylogeographic history, showing the mito-nuclear discordance of an endemic groundwater amphipod from Iceland, *Crangonyx islandicus*, which is the only metazoan known to have survived the Pleistocene beneath the glaciers. Previous studies based on three DNA markers documented a mitochondrial scenario with the main divergence occurring between populations in northern Iceland and an ITS scenario with the main divergence between the south and north. We used double digest restriction-site-associated DNA sequencing (ddRADseq) to clarify this mito-nuclear discordance by applying several statistical methods while estimating the sensitivity to different analytical approaches (data-type, differentiation indices and base call uncertainty). A majority of nuclear markers and methods support the ITS divergence. Nevertheless, a more complex scenario emerges, possibly involving introgression led by male-biased dispersal among northern locations or mitochondrial capture, which may have been further strengthened by natural selection.

**Keywords:** introgression; incomplete lineage sorting; selection; subglacial refugia; populations; genomics; groundwater; RADseq



## 1. Introduction

An understanding of the evolutionary histories of species, including geographic population structure, degree of connectivity/or isolation, past demographics or migration routes, offers crucial knowledge about species and is the main goal of phylogeography [1–4]. Phylogeographic studies have traditionally relied on small fragments of mitochondrial DNA (mtDNA) [2,3]. The predominance of mitochondrial markers was predicated by a well-developed procedure to obtain data and by properties that allowed the inference of relationships between closely related taxa such as fast mutation rate, low or absent recombination rate, small population size and quasi-neutral evolution [2,5], but see [3]. However, the mitochondrial genome is only a single marker that can have a singular evolutionary history [1,6]. The analysis of variation within the nuclear genome is thus needed in order to

obtain a better estimate of the history of the populations studied [1,6]. Such studies have in some cases revealed divergent phylogeographic scenarios than inferred from mtDNA, i.e., mito-nuclear discordance conundrums [1,6], and raised questions about the inferences of the underlying evolutionary processes [1,6,7]. Two studies in the highly dynamic Icelandic aquatic ecosystem showed that an endemic groundwater amphipod species, *Crangonyx islandicus* Svavarsson and Kristjánsson, 2006, presented a mito-nuclear discordant phylogeographic pattern interrogating its evolutionary history and population structure [8,9]. As the observed discordance was limited to a single nuclear marker [9], sampling a larger proportion of the nuclear genome is warranted to assess the robustness of the mito-nuclear discordance and the evolutionary history of *C. islandicus*. However, it poses additional analytical challenges when considering a large diversity of nuclear markers.

Mito-nuclear discordance has been found in several species and can be driven both by stochastic and introgression processes [1,6,7,10–12]. Stochasticity can lead to incomplete lineage sorting (ILS) among genes due to genetic drift [1], which implies the persistence of ancestral polymorphism and in different genealogies among genes [1,6,7]. Different genealogies can, in addition, be maintained by different modes of inheritance and ploidy (e.g., bi-parental inheritance for nuclear DNA vs. maternal inheritance for mitochondrial DNA). Incomplete lineage sorting usually does not leave a particular pattern, and frequencies among different patterns should be homogeneous. On the contrary, introgression implies hybridization among individuals coming from distinct isolated populations through secondary contact allowed by the removal of the dispersal barrier. Introgression should be more prominent among taxa with closer relationships than among distantly related species [13]. Mito-nuclear discordance resulting from introgression can be driven by adaptation or sex-biased mechanisms such as sex-biased dispersal, hybrid survival and mate choice [6,7]. Purifying or balancing selection, augmented by selection on linked sites due to reduced or no recombination, may in addition lead to deviations in genetic patterns among different genomic regions such as in mitochondria, sex- and autosomal chromosomes (e.g., [14]), and such impacts can be stronger in small populations where mutations with weak fitness effects can accumulate [15]. As mito-nuclear discordance can strongly affect the inferences about the underlying evolutionary processes [1,6,7], implications for conservation measures may need to be revisited as the population structure, gene flow and effective population size inferred from a single or few markers may not represent the divergence history among populations but a singular pattern [16]. Revising mitochondrial phylogeographic patterns with nuclear DNA input is thus crucial to clarify the relationships of closely related taxa and the population processes (drift, mutation, migration, selection) that shape genetic diversity as a whole.

Abundant polymorphic nuclear markers can now be obtained for non-model organisms through reduced representation genome sequencing methods such as restriction-site-associated DNA sequencing (RADseq) [17,18]. Restriction-site-associated DNA sequencing has been successfully used to infer recent population structure [19,20], population history [21,22], resolve enigmatic phylogenetic relationships between species [23–25] and to assess the extent of hybridization and introgression between species [26]. However, two points need to be considered when analyzing RADseq data to infer phylogeographic scenarios. First, RADseq loci present a large diversity of markers with different modes of inheritance (mitochondrial vs. nuclear) and differences within the genome such as variation in mutation rates (e.g., in conserved coded regions and repetitive neutral regions), recombination rates and selection pressure [27]. Consequently, the genetic diversity among loci may be highly variable, especially when the full DNA sequence of different loci is considered instead of unlinked single nucleotide polymorphisms (SNPs) per locus. Statistics that are used to infer putative population structure or population parameters can be differently sensitive to this variability in genetic diversity [28–31]. For instance, the differentiation indices F-statistics [32] and Gst [33], contrary to the D statistic [29], are sensitive to high levels of heterozygosity and can thus be misleading to reveal population structure, but still offer a good estimate of the demographic history [34]. Care is thus

needed when interpreting commonly reported population genetic summary statistics as the type of genetic information retained (full sequence, haplotype or unlinked SNPs) may amplify variability in genetic diversity among loci. Second, there is a trade-off between the number of individuals sampled and the accuracy of the genotype assigned by the sequencing read depth at each locus [35–37]. For a given sequencing effort, higher sample size increases estimates of allele frequencies within a population and the probability of detecting rare alleles, but at the cost of decreasing sequencing read depth across loci, which may not be sufficient to remove sequencing errors and potentially biasing downstream summary statistics [38–40]. Recently, several methods have been developed to overcome this issue taking into account the base call uncertainty [38,39,41–44]. Despite the fact that these methods are still in their infancy, they offer the opportunity to evaluate the robustness of the summary statistics using classic base calling for low-coverage datasets [44].

In this study, we revisit the phylogeographic history, showing the mito-nuclear discordance of *C. islandicus* using mitochondrial and RADseq data, while estimating the sensitivity of different analytical approaches (data-type, differentiation indices and base call uncertainty). *Crangonyx islandicus* is one of the two groundwater amphipod species endemic to Iceland that are found in springs in lava fields throughout the volcanic zone of the country [8,45,46]. It is the only metazoan species known to have survived beneath the glaciers during the Pleistocene, likely in fissures along the tectonic plate boundary in Iceland [8] (Figure 1a). A phylogeographic study based on its 16S and COI mitochondrial genes revealed a clear pattern of population divergence of five mitochondrial monophyletic clades, hereafter referred to as the mitochondrial scenario (Figure 1b) (clades AA', BC, D, E and F). The differentiation of the clades followed geographical separation within Iceland; clades AA', BC and D in southern Iceland diverged from each other from 0.4 to approximately 1 Myr ago, from E about 1.3 Myr ago and clade F, located in northeastern Iceland, presenting a putative cryptic species, diverged approximately 4.8 Myr ago from the others [8]. However, the variation in the nuclear internal transcribed spacer regions (ITS) 1 and 2 showed a different phylogeographic pattern potentially driven both by concerted and divergent evolution, hereafter referred to as the ITS scenario (Figure 1b) [9]. Contrary to the mitochondrial markers, ITS1 showed the oldest split between locations from northern Iceland (E and F) and those from southern Iceland, which share a common duplicated ITS1 region of 269 base pairs. Contrary to the mitochondrial DNA, ITS do not support the presence of cryptic species [9]. Further nuclear information is needed to evaluate the discrepancy between the patterns obtained with these two markers.

Here, we used the mitochondrial and double digest RADseq (ddRADseq) sequence data of the individuals sampled at four locations belonging to three mitochondrial clades (AA', E and F) previously identified by Kornobis et al. [8] to revisit the phylogeography of *C. islandicus* and test the likelihood of the mitochondrial vs. ITS scenario. First, to evaluate the mitochondrial scenario proposed by Kornobis et al. [8], we performed a Bayesian structured coalescent analysis [47] with the subset of three clades to reconstruct the phylogeographic history of the mitochondrial DNA. Second, we used ddRADseq data to test population structure using different data types (SNP, haplotypes, sequences), differentiation indices, including indices taking into account the base call uncertainty and multivariate and Bayesian multi-coalescence analyses. Finally, using ddRADseq data we evaluated the level of incongruence among loci, and tested the amount of admixture between sampling locations caused by introgression or ILS.

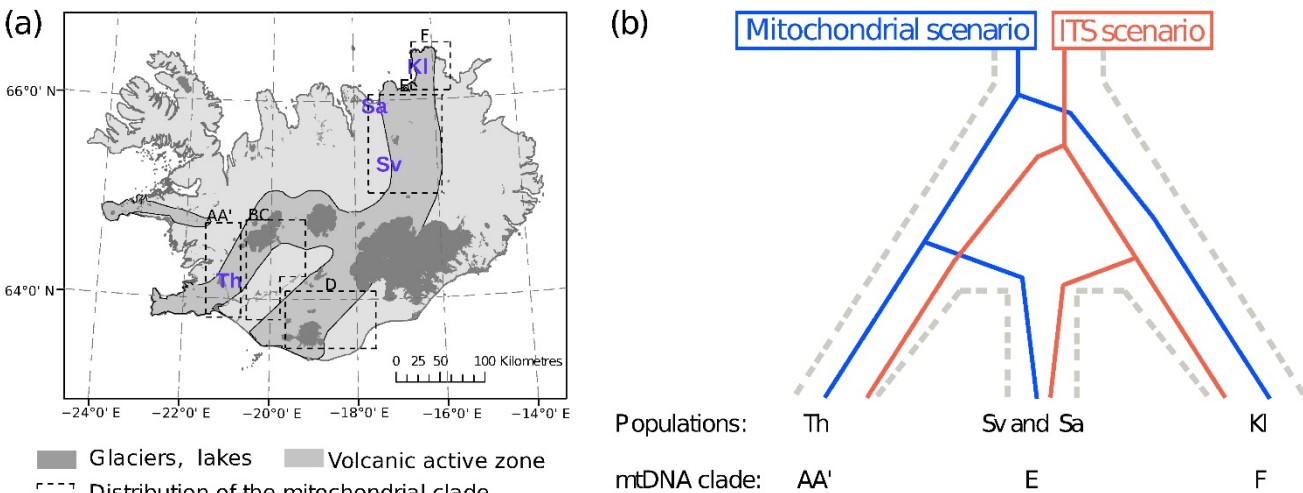

**Figure 1.** Sampling sites, distribution of mitochondrial DNA (mtDNA) lineages and comparison of divergence in mtDNA and in the internal transcribed spacer region (ITS). (**a**) Map of the sampling sites (in purple) and schematic distribution, denoted with squares and capital letters, of the mitochondrial clades (AA', BC, D, E and F) previously defined [8]. (**b**) Two alternative scenarios of the population relationships proposed in the literature [8,9] showing mito-nuclear discordance. The grey dotted line represents the undefined "species tree". Th: Lake Thingvallavatn, Sa: Sandur Adaldal, Sv: Lake Svartárvatn, Kl: Klapparós.

## 2. Materials and Methods

### 2.1. Sampling

Fifty-nine individuals of *C. islandicus* were sampled in four springs in 2013, representing three distinct mitochondrial clades [8]: clade AA' in Lake Thingvallavatn (Th: southern Iceland), clade E at Sandur Adaldal (Sa) and in Lake Svartárvatn (Sv) (northern Iceland) and clade F in Klapparós (Kl: northeastern Iceland) (Figure 1a, Supplementary Materials Table S1.1). We used dip nets after applying electricity with electric fishing gear to sample specimens near groundwater springs. Specimens were fixed in 96% ethanol and stored at −20 °C, or frozen with dry ice and stored at −80 °C.

### 2.2. Molecular Techniques

Total genomic DNA was extracted from the fifty-nine individuals using a standard phenol-chloroform protocol [48].

Mitochondrial DNA: To validate the mitochondrial pattern observed for these locations by Kornobis et al. [8], 21 out of the 59 individuals were newly sequenced for the CO1 and 16S genes in this study and these sequences were completed by 37 individuals previously sampled in the immediate surroundings of these four locations [8] for a total of 58 individuals used in the mitochondrial analysis (see Supplementary Materials, Table S1.2). For PCR amplifications of the CO1 and 16S genes for 21 new individuals, we followed the same protocol as described in Kornobis et al. [8], and the sequencing was performed in both directions using the Sanger method on a Genetic Analyser (3500xL Applied Biosystem, Waltham, MA, USA).

Restriction-site-associated DNA sequencing: A double digest restriction-site associated DNA sequencing (ddRADseq) library was constructed from 59 individuals, using modified protocols from [49,50]. Total genomic DNA (100–500 ng) was sequentially digested using the restriction endonucleases Sau3AI (1U) and ApeKI (2U), respectively, each for 4 h at the manufacturer's (NEB) recommended temperatures in NEB Buffer 4. Digested DNA (100 ng) was ligated to adapters (sequences in [51]) containing unique combinatorial barcodes (16 unique 5 bp barcodes for ApeKI adapters and 5 unique 6 bp barcodes for Sau3AI adapters) for each individual (barcode and adapter sequences in Supplementary

Materials S2) using T4 DNA ligase (NEB) in supplied buffer at 21 °C for 4 h. Ligation reactions contained a 6:1 molar excess of adapter to fragmented DNA, calculated using the mean fragment size determined from an agarose gel. Ligated DNA was pooled and purified using magnetic beads (Macherey-Nagel NGS clean-up and size selection) following the manufacturer's protocol. Size selection of ligated DNA fragments was performed on a Pippin Prep (Sage Science, Beverly, MA, USA) with 2% ethidium-free agarose gels and external size standard. The narrow range setting included a mean fragment size of 350 bp ± 18 bp. The eluate was split among eight PCR reactions and amplified using the primers and PCR conditions as described in Elshire et al. [51]. Each PCR reaction had a total volume of 25 μL containing 1× OneTaq Master Mix with Standard Buffer (NEB), 0.5 mM each primer and 8 μL template DNA. Polymerase chain reaction products were pooled and purified by magnetic beads before quantification using a SYBR Gold fluorometric assay (protocol in Supplementary Materials S3). The library was prepared for sequencing following the manufacturer's instructions with a final concentration of 38 nM. The library was sequenced on an Illumina HiSeq2500 using the Illumina TruSeq kit (2 × 125 bp). To increase coverage of the data, the library was resequenced twice on an Illumina MiSeq2000 using the MiSeq Reagent Kit v2 (2 × 150 bp).

### 2.3. Bioinformatic Pipeline

2.3.1. Single Nucleotide Polymorphisms, Haplotypes and Sequence Datasets Base-Called with pyRAD

The process_radtag.pl command in STACKS v1.3 program [50] was used to demultiplex the paired-end libraries from ddRadseq. Reads with uncalled and low quality (phred score < 10) bases were excluded, the remaining reads trimmed to 100 bp and barcodes and adapters removed. The reads of the HiSeq run and the two MiSeq runs were assembled. The reads were assembled de novo into loci using pyRAD v1.3.1 [52], as the alignment clustering approach allows for the presence of indels to improve the identification of homology among divergent taxa. The restriction recognition overhang sites were removed. Low quality converted base calls below a score of 20 in pyRAD were considered as N and reads with more than 10 Ns were removed. A threshold of 80% was used to cluster the reads within individuals [25]. Heterozygosity and error rate were estimated in pyRAD using the maximum likelihood formula proposed by Lynch [53]. We used a statistical base call with a minimum depth coverage of five and then, due to low-coverage data, we generated two additional datasets using a consensus base call with a minimum depth coverage of three (C3) and two (C2) to retrieve additional loci. We removed the consensus sequences with more than 10 Ns. To exclude potential paralogs, the consensus sequences with more than two alleles after error correction and 10 heterozygous sites were removed (step 5 in pyRAD). Then, the consensus sequences were clustered across individuals at 80% similarity [24] and aligned (step 6 in pyRAD). Finally, we retained all the loci present at least in 50% of the individuals (*n* = 30) to generate the output formats of the C2 and C3 datasets (step 7 in pyRAD).

Single nucleotide polymorphism analyses were performed using one SNP per locus. The SNPs were selected either randomly (C2SNPr, C3SNPr, see Table 1) or selecting the most informative SNP ((C2SNPb, C3SNPb) as the one with the lowest number of missing data (individuals) and in the case of equal amount of missing data one was chosen randomly). The SNP selections were performed in R [54] from the .vcf output file provided by pyRAD. Haplotype analyses were performed using the information of all the SNPs present in a locus. We used the .alleles output file provided by pyRAD to generate fasta alignments for each locus. Then, we generated the haplotype files in Fstat formats removing the sequences with more than 25% ambiguous position (i.e., second read missing) and considering indel as a 5th state (Cov2H5th, Cov3H5th) or as missing data (Cov2Hmis, Cov3Hmis) using the haplotype function of the R package haplotypes [55]. Finally, all the haplotype datasets were also derived in their sequence equivalent datasets considering nucleotide information (Table 1).

To test the influence of low-frequency alleles, potential remaining paralogs and loci detected under selection, we considered a "brute" dataset and a "clean" dataset, where we filtered out the loci in low frequency, considered as potential paralogs and under selection. We removed all the loci with a minimum allele frequency (MAF) below 5% [19,56] and those with an observed heterozygosity above 0.5, which can be considered as potential paralogs [19,57]. To remove loci detected under selection, we ran Bayescan v2.01 [58], considered as one of the most robust programs, using the differentiation approach [59]. To decrease the high false discovery rate, we followed Lotterhoos and Whitlock [60] using a prior odd = 10,000. For each SNP and haplotypes dataset, we performed two Bayescan runs using 50,000 MCMC generations preceded by 20 pilot runs of 5000 generations. The program Blastn v2.4.0 in NCBI (https://blast.ncbi.nlm.nih.gov/Blast.cgi?PROGRAM= blastn&PAGE_TYPE=BlastSearch&LINK_LOC=blasthome (accessed on 16 March 2016)) was used to infer the role of loci putatively under selection through search in GenBank. Loci identified to be under selection were further analyzed by designing primers in sequences under selection and by amplifying and sequencing from up to 79 individuals per loci from 14 sites (see Supplementary Materials S4 for additional details) from the species range [8]. The population patterns of these markers were then compared to the overall pattern obtained from this study and previous studies [8,9].

### 2.3.2. Genotype Likelihood Datasets with ANGSD

As low-coverage datasets are prone to base call uncertainty, which can bias downstream analyses [39,40], we applied recent methods that by-pass the base call step and estimate the genotype likelihood [38] to compute almost unbiased summary statistics such as Fst-like [39]. Such an approach implemented in ANGSD [43] and other related software [41] requires converting the original fastq files into BAM files. As no reference genome was available for *C. islandicus* or its closely related taxa, we generated two mock reference genomes for each dataset (cov2 and cov3) using a consensus sequence with a threshold of 95% similarity among all the individuals of a concatenated sequence of all the selected loci assembled by PyRAD (see previous paragraph). The demultiplexed fastq files of each individual were aligned separately against these mock reference genomes in Bowtie2 [61] and only the aligned paired-end reads were retained and converted into SAM files. SAMTOOLS [62] was used to sort the reads and convert the files in BAM format for the two datasets (cov2GL and cov3GL). ANGSD v0.9.1 was used to perform quality filtering of the reads. Bad reads with SAM score > 255 (-remove_bad 1) and reads with multiple best hits (-unique_only 1) were removed with a minimum mapping quality of 30 (-minMapQ 30) and a minimum base quality of 20 (-minQ 20). Sites likely to be polymorphic with a *p*-value less than $10^{-6}$ (-SNP_pval 1e-6) and with a minimum coverage of 2 and a maximum coverage of 20 for at least 50% of the individuals were retained. To test the effect of low-frequency sites, we also considered a "brute" dataset and a "clean" dataset, where the sites with a MAF < 0.05 were removed. For all these datasets, the genotype likelihood function provided by SAMTOOLS [62] and implemented in ANGSD (-GL 1) was applied.

### 2.4. Mitochondrial Analysis

All the CO1 and 16S sequences were aligned by Clustalo, implemented in Seaview v.4.2.12 [63] and checked by eye. A Bayesian structured coalescent tree was reconstructed using MultiTypeTree (MTT) [47] in BEAST2 v. 2.3.1 [64] to estimate the tree topology, robustness, divergence time, migration rate among the three mitochondrial clades defined by Kornobis et al. [8] (clades AA', E and F) as well as to assess their effective population size. We estimated the best substitution model for the CO1 (TN93) and 16S (HKY+I) genes according to the Akaike criterion corrected for a small sample size (AICc) with MrAIC.pl v1.4.6 [65]. Two MTT runs were performed considering independent substitution and clock models for CO1 and 16S but with a fixed tree topology because both belong to the same linkage group. To estimate the divergence date, we used the same set-up as in

Kornobis et al. [8], applying a strict molecular clock, divergence rates of 1.4–2.6% per million years for CO1 and 0.53–2.2% for 16S and using a uniform distribution prior. For the substitution models, we used the default priors, while for the population size and migration rate prior we used a Lognormal distribution (mean = 0, sd = 2). The two independent Monte Carlo Markov Chains (MCMCs) were run for 200 million generations, sampled every 20,000 generations. The first 10% of each chain was discarded as burn-in, and then both chains were combined using LogCombiner. Finally, we checked the convergence of the combined chain (ESS > 200) with TRACER V1.6 [66] and extracted the maximum clade credibility tree (MCCT) using TreeAnnotator, which was displayed with FigTree v1.4.2 (available at http://tree.bio.ed.ac.uk/software/figtree/, accessed on 26 March 2016).

**Table 1.** Datasets used in the present study and the number of markers (i.e., SNPs or loci) retained after filtering (Nb. Mark.). Dataset Name: abbreviations of the name of datasets, Data Type: the type of genetic data involved for each locus (i.e., SNP, haplotype or DNA sequence), Min. Cov: the minimal coverage retained for a locus, Cl. Data: using a brute (br) dataset coming from pyRAD or a cleaned (cl) dataset removing loci with a minimum allele frequency (MAF) below 5%, or with an observed heterozygosity above 0.5 or detected under selection by Bayescan [58]. Nb. Mark: number of DNA markers retained in the analysis.

| Dataset Name | Data Type | Min. Cov. | Cl. Data | Nb. Mark. |
|---|---|---|---|---|
| C2SNPr_br | SNP (random SNP) | 2 | br | 326 |
| C2SNPb_br | SNP (best SNP) | 2 | br | 326 |
| C2SNPr_cl | SNP (random SNP) | 2 | cl | 230 |
| C2SNPb_cl | SNP (best SNP) | 2 | cl | 196 |
| C2Hmiss_br | Haplotype (indel missing) | 2 | br | 295 |
| C2H5th_br | Haplotype (indel 5th state) | 2 | br | 313 |
| C2Hmiss_cl | Haplotype (indel missing) | 2 | cl | 202 |
| C2H5th_cl | Haplotype (indel 5th state) | 2 | cl | 228 |
| C3SNPr_br | SNP (random SNP) | 3 | br | 108 |
| C3SNPb_br | SNP (best SNP) | 3 | br | 108 |
| C3SNPr_cl | SNP (random SNP) | 3 | cl | 90 |
| C3SNPb_cl | SNP (best SNP) | 3 | cl | 73 |
| C3Hmiss_br | Haplotype (indel missing) | 3 | br | 103 |
| C3H5th_br | Haplotype (indel 5th state) | 3 | br | 109 |
| C3Hmiss_cl | Haplotype (indel missing) | 3 | cl | 71 |
| C3H5th_cl | Haplotype (indel 5th state) | 3 | cl | 83 |
| C2Seqmiss_br | Sequences the same as C2Hmiss_br | 2 | br | 295 |
| C2Seqmiss_cl | Sequences the same as C2Hmiss_cl | 2 | cl | 202 |
| C2Seq5th_br | Sequences the same as C2H5th_br | 2 | br | 313 |
| C2Seq5th_cl | Sequences the same as C2H5th_cl | 2 | cl | 228 |
| C3Seqmiss_br | Sequences the same as C3Hmiss_br | 3 | br | 103 |
| C3Seqmiss_cl | Sequences the same as C3Hmiss_cl | 3 | cl | 71 |
| C3Seq5th_br | Sequences the same as C3H5th_br | 3 | br | 109 |
| C3Seq5th_cl | Sequences the same as C3H5th_cl | 3 | cl | 83 |
| C2GLr_br | Genotype likelihood | 2 | br | 310 [b] |
| C2GLr_cl | Genotype likelihood | 2 | Cl [a] | 300 [b] |
| C3GLr_br | Genotype likelihood | 3 | br | 103 [b] |
| C3GLr_cl | Genotype likelihood | 3 | Cl [a] | 103 [b] |

[a] Loci with a MAF below 5% were removed. [b] Maximum number of markers used in pairwise comparisons.

### 2.5. Restriction-Site-Associated DNA Sequencing Analysis

2.5.1. Population Differentiation Estimates among Sampling Locations

Pairwise population differentiation was estimated using five different metrics: $G_{ST}$, $G''_{ST}$ [30], $D_{ST}$ [29], $\Phi_{ST}$ and $F_{ST}$-like statistics based on genotype likelihood [39]. $G_{ST}$, $G''_{ST}$ and $D_{ST}$ were estimated for SNP and haplotype datasets, and $\Phi_{ST}$ was estimated on the sequence datasets. We use $G_{ST}$, $G''_{ST}$ and $D_{ST}$ due to their different sensitivity to the mutation rate, level of heterozygosity and population size [30,34]. $G''_{ST}$, the standardized

measure of $G_{ST}$ taking into account bias of sample size, and $D_{ST}$ are less sensitive to the level of heterozygosity (genetic diversity). $D_{ST}$ [29] is more sensitive to the variation of mutation rate between loci but it is considered as a better measure of differentiation in a wide range of circumstances because it is less sensitive to the population size [29–31,34]. To test the statistical significance of the differentiation among populations, we used Fisher's exact test, implemented in the mmod R package [67] using 1000 permutations for each locus. Then, the *p*-values of each locus were combined following Fisher's method [68] and corrected using Holm's formula for multiple testing [69]. We also computed the pairwise genetic distances between populations by taking differences between sequences into account ($\Phi_{ST}$), using K80 (Kimura 2 parameters) and TN93 [70] substitution models. As both models provided similar results ($R^2 = 0.99$), only the K80 model $\Phi_{ST}$s were retained. Statistical significance of the pairwise $\Phi_{ST}$s was evaluated using 1000 permutations for each locus and the *p*-values were combined using Fisher's method [68] and corrected for multiple testing according to Holm's formula [69]. Pairwise $G_{ST}$, $G''_{ST}$ and $D_{ST}$s were estimated using the mmod R package, while $\Phi_{ST}$ was computed with the pegas R package [71]. The base call uncertainty caused by the low-coverage dataset was taken into account by computing a Fst-like statistic proposed by Fumagalli et al. [39], based on genotype likelihood implemented in ANGSD using realSFS tools.

The differences between various pairwise differentiation indices and different datasets in Table 1 were summarized by calculating distances and visualized with a metric multidimensional scaling approach in R. The distances were calculated as 1-*r*, where *r* is the Pearson correlation coefficient based on the pairwise distances among populations computed among indices/datasets.

### 2.5.2. Congruence Tests among Loci

An empirical approach was used to estimate the proportion of loci supporting the mitochondrial scenario (Klapparós, considered as the most divergent population, was used as an outgroup) vs. the ITS scenario (Lake Thingvallavatn as an outgroup) using $G_{ST}$, $G''_{ST}$, $D_{ST}$ and $\Phi_{ST}$ as differentiation measure among sampling locations. The difference in the extent of genetic differentiation between the outgroup and the other populations obtained for the two scenarios was tested using a non-parametric Wilcoxon test in R.

### 2.5.3. Population Structure and Relationship among Groups

Population structure was further inferred with both SNPs and haplotype datasets using the discriminant analysis of principal components (DAPC [72]) implemented in the adegenet v.2.0.1 R package [73]. The DAPCs were performed both considering the best clustering scheme inferred by the sequential K-means approach and using the sampling sites as a prior. The choice of the optimal number of clusters (K), within a range of 2 to 10, was performed according to the lowest score of the BIC criteria. The optimal number of retained principal components was performed using a cross validation procedure [72].

The statistical support of the number of independent lineages (or populations) and their phylogenetic relationships was estimated by performing a Bayesian multi-coalescence analysis using unlinked bi-allelic SNP with SNAPP [74] with a Bayes Factors Delimitation (BDF*) procedure [75,76], implemented in BEAST2. SNAPP estimates the species tree using unlinked bi-allellic SNPs while by-passing the estimate of multiple gene trees. It allows estimating the statistical support of the tree topology and posterior probability of the present and ancestral effective population size assuming constant population size for each branch of the tree and sudden changes [74]. For computational tractability, five individuals were selected per location with the least number of missing bi-allelic SNPs for the datasets with a minimum coverage of 2 and 3, respectively, and after removing loci in low frequency and potential paralogs. Three models were compared, M1: the four sampling locations as distinct entities [(Sv,Sa,Th,Kl)], M2: the mitochondrial scenario with three distinct entities [(((Sv,Sa),Th), Kl)] and M3: the ITS scenario with two distinct entities (the populations in the north vs. Thingvallvatn in the south [((Sv,Sa,Kl), Th)]. For each model, two MCMCs of

2,000,000 iterations sampled every 100 iterations were performed. The first 10% of each chain was discarded and the remainder of both chains was combined using LogCombiner. The convergence of the parameters was checked with TRACER (ESS > 200).

The statistical supports of the three models were estimated with the Bayes Factors (BF), using a path-sampling approach to estimate the marginal likelihoods [76]. Two independent runs were conducted for each model with 20 steps of path sampling with 100,000 MCMC iterations per step. We sampled the chain every 100 iterations after discarding the first 1000 steps as pre-burn-in and then the first 10% as burn-in. The average marginal likelihood (*ML*) of the two runs of each model was calculated and the formula $BF = 2*(ML_{model1} - ML_{model2})$ was used to compare the two models. According to the scale of Kass and Raftery [77], the support for model 1 is considered to be strong for BF > 2, very strong for BF > 6 and decisive for BF > 10.

### 2.5.4. Introgression Analysis

The four-taxon D statistic, also called ABBA-BABA test [78,79], was used to disentangle the contribution of ILS and introgression (admixture) caused by ancestral hybridization among the four putative populations. If "A" is an outgroup/ancestral allele and "B" a derived allele, the D statistic represents the occurrence of two discordant patterns, ABBA and BABA, in a four-taxon tree topology [(((P1,P2),P3),O)], with P1, P2 and P3 as clades under investigation and O as an outgroup. Under ILS caused by stochastic sorting of ancestral polymorphism, frequencies of both patterns are expected to be similar, while an excess of one of those patterns may arise if introgression occurs between P3 and P2 or P1 [24,79]. D is positive when P2 exchanges with P3 and D is negative when P1 exchanges with P3. We used the SNP frequency of heterozygous site to compute the D statistic considering two alternative scenarios: the mitochondrial scenario tests the introgression between Thingvallavatn (P3) and Svartárvatn (P1) or Sandur (P2) using Klapparós as an outgroup, and the ITS scenario tests the introgression between Klapparós (P3) and Svartárvatn (P1) or Sandur (P2) using Thingvallavatn as an outgroup. Switching the position of Svartárvatn and Sandur did not qualitatively affect the results. To obtain the most complete data matrix, we selected the five most informative individuals per population (maximizing the locus coverage among populations) for two datasets (with a minimal coverage of 2 and 3, respectively) and after removing low-frequency data (Cov2Hmiss_cl and, Cov3Hmiss_cl datasets). For the ingroup taxa, we selected randomly one individual for each of P1, P2 and P3 populations among the five individuals retained per population, while pooling the five individuals of the outgroup to estimate the SNP frequency [26]. For each of these combinations of individuals (i.e., 125 replicates in total), we computed the D statistic. To assess the significance of the D statistic over replicates, we followed the procedure of Eaton et al. [26]. For each replicate, we performed 200 bootstrap iterations over loci with replacement to estimate the standard deviation of the D statistic and convert the observed D statistic as a Z-score. The significance of the Z-score was assessed by a conservative *p*-value at alpha = 0.01 after a Holm correction for multiple testing over replicates. A D statistic with an absolute Z-score value above 3 is generally considered as significant.

## 3. Results

### 3.1. Mitochondrial Results

The MultiTypeTree analysis based on the two mitochondrial genes (CO1 and 16S), including twenty-one new individuals, perfectly supports the previously described mitochondrial scenario [8]. The MCCT showed four well-supported monophyletic groups (node pp = 0.99–1, Figure 2a). The individuals from Klapparós belong to the clade F, the individuals of the two northern locations (Sandur and Lake Svartárvatn) belong without sub-geographical structure to the clade E, and the individuals sampled in Lake Thingvallavatn belong to the clade AA'. The confidence interval of the date of divergence between the clade F (95% HPD = 2.3–4.8 Myr) and all remaining clades (95% HPD = 0.7–1.8 Myr)

did not overlap and supports a clear ancestral divergence of the mitochondrial genes between the northeastern population (Clade F) and a clade formed by the ancestors of the populations in the north (clade E) and the south (clades A and A', Figure 2a). There was no evidence of recent mitochondrial gene flow among the mitochondrial clades, i.e., the confidence intervals of the posterior distribution of the migration rates always overlapped 0. Assuming a constant mutation rate for all the mitochondrial clades, the posterior distributions of $\theta$ showed that the Thingvallavatn population (Clade A-A') tends to have a higher effective population size ($\theta_{median}$ = 0.49, 95% HPD = 0.23–0.87) than that from Klapparós ($\theta_{median}$ = 0.14, 95% HPD = 0.03–0.35) and the northern population (both samples of the clade E, Sandur and Svartárvatn, were considered as one population according to the absence of geographical structure, $\theta_{median}$ = 0.07, 95% HPD = 0.01–0.18, Figure 2b).

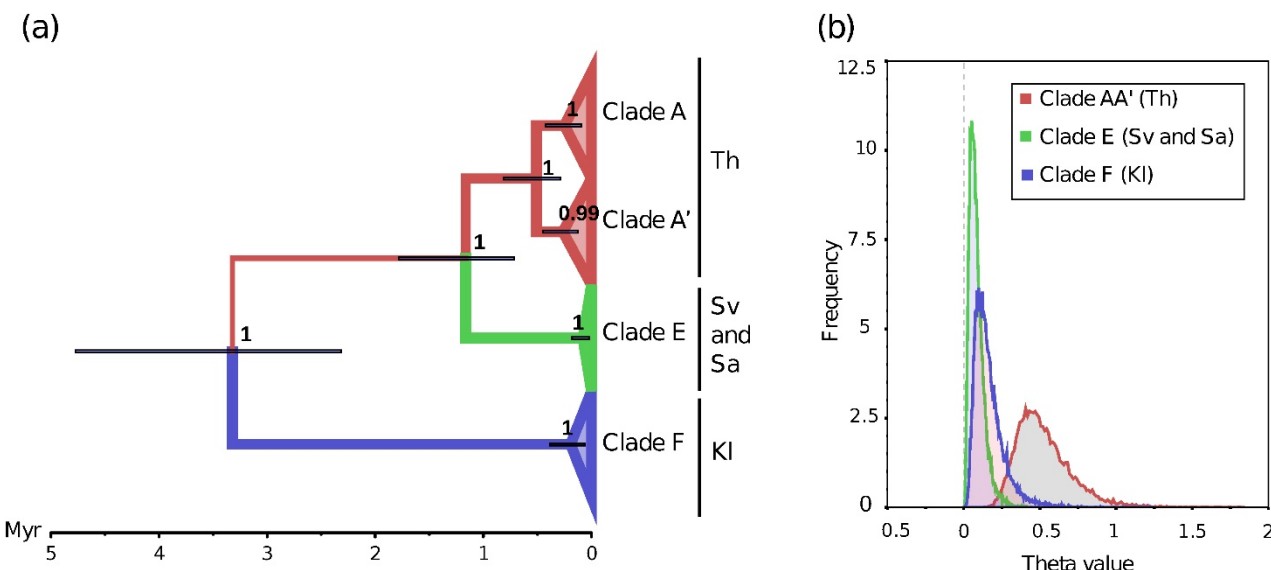

**Figure 2.** Divergence of mitochondrial genotypes of *Crangonyx islandicus* from four geographic locations. (**a**) Maximum clade credibility tree (MCCT) and (**b**) posterior distribution of $\theta$ ($\theta = Ne*\mu$) parameters inferred by MultiTypeTree [47] in BEAST2 for four locations of *Crangonyx islandicus* using CO1 and 16S mitochondrial genes belonging to three mitochondrial clades, previously identified by [8]. Color branches of the MCCT represent the most likely localization of the ancestors of the four populations inferred by MultiTypeTree. Theta = $Ne*\mu$, Ne = effective population size, $\mu$ = mutation rate. Th: Lake Thingvallavatn, Sv: Lake Svartárvatn, Kl: Klapparós, Sa: Sandur.

### 3.2. Restriction-Site-Associated DNA Sequencing Results

#### 3.2.1. Description of the Datasets

The difference in the minimal coverage used, from two to three, had a major effect on the total number of de novo loci and SNPs retained in the analysis by pyRAD shifting from 335 to 123 loci, respectively, of which 323 and 108 were polymorphic, respectively. The coverage cut-off also reduced the dataset from 3539 SNPs with two to 835 SNPs with three. The haplotype datasets contained more alleles when indels were considered as 5th state compared to when they were considered as missing data (C2, missing = 4.5 ± 3.3; 5th = 6.5 ± 4.9; paired Wilcoxon test $p < 0.001$: C3, missing = 3.8 ± 3.2; 5th = 5.5 ± 4.7; paired Wilcoxon test $p < 0.001$). No unlinked SNPs were identified as putatively under selection by Bayescan, while haplotype datasets revealed seven loci under putative selection using a minimum coverage of two, regardless of whether the indels are considered as missing data or not. For a minimum coverage of three, we detected five and six loci under putative selection when indels were considered as missing data or as 5th state, respectively. Among the loci detected as putatively under selection, the Blastn approach was able to retrieve three annotated DNA fragments with an E-value statistic below 0.001: a histone H3 nuclear gene, a mitochondrial NADH dehydrogenase subunit 5 and another mitochondrial fragment

overlapping the genes ND2 and tRNA-Trp (Supplementary Materials Table S5.1). The NADH dehydrogenase subunit 5 was found only with the dataset with a minimal coverage of two. The proportion of loci in low frequency (with MAF < 0.05), under putative selection and with observed heterozygosity above 0.5 was $29.6 \pm 9.7\%$ for the SNPs and $28.4 \pm 3.6\%$ for the haplotypes. These data were removed when considering the "clean" datasets.

### 3.2.2. Differentiation among Populations

A large majority (i.e., 92%) of the datasets and the differentiation metrics supported the ITS scenario (Figure 3). The $\Phi_{ST}$, in particular, was insensitive to the datasets used and supported systematically the ITS scenario (Figure 3). For all datasets, the significance of the pairwise $G_{ST}$, $G''_{ST}$ and $D_{ST}$ between populations was always supported ($p < 0.001$), even after $p$-value correction for multiple testing among populations. However, all the pairwise $\Phi_{ST}$ between the southern (Thingvallavatn) and the northern populations (Sandur, Svartárvatn and Klapparós) were significant at $\alpha = 0.05$ but not among the northern populations (except between Sandur and Svartárvatn, with a minimum coverage of three, Table S5.2).

All unlinked SNP datasets supported the ITS scenario regardless of the differentiation metrics, t coverage or the presence/absence of the loci in low-frequency or under selection (Figure 3, Table S5.2). Similarly, regardless of the dataset used, the $F_{ST}$-like estimates based on genotype likelihood taking into account the base call uncertainty always supported the ITS scenario. However, minimal coverage had a stronger effect on the differentiation among populations than the effect of removing the loci under low frequency at a given coverage (Figure 3). Furthermore, all haplotype datasets with a minimal coverage of three supported the ITS scenario regardless of the differentiation metrics used. However, with a coverage of two, considering the presence of indels as informative or not, removing loci in low frequency and under selection affected the differentiation metrics variably, as well as the inferred scenario. Two approaches showed opposite results. First, $G_{ST}$, $G''_{ST}$ and $D_{ST}$ supported the ITS scenario when indels were considered as missing data and all the loci were retained, while in the second approach, considering indels as a 5th state and removing the loci in low frequency and under selection, these three metrics supported the mitochondrial scenario. For the latter approach, 68% of the loci removed supported the ITS scenario. Finally, two haplotype datasets (C2Hmiss_cl and C2H5th_br) showed discordant results when using different differentiation metrics, with the $G_{ST}$ supporting the mitochondrial scenario, while the $G''_{ST}$ and $D_{ST}$ supported the ITS scenario. Such discrepancies between the $G_{ST}$ and the two other differentiation metrics might be explained by the greater sensitivity of $G_{ST}$ to the significant increase in the number of alleles and the heterozygosity within subpopulation (Hs) for all the loci supporting the discordant scenario compared to all the loci supporting the concordant scenario (Cov2H5th_br: nb loci discordant = 22, Wilcoxon test $P_{Hs(global)} = 0.024$, $P_{nbAlleles} < 0.001$; Cov2Hmiss_cl: nb loci discordant = 7, Wilcoxon test $P_{Hs(global)} = 0.005$, $P_{nbAlleles} = 0.012$).

Globally, and for each pairwise population comparison, the $G''_{ST}$ showed a higher absolute differentiation value and $D_{ST}$ showed the lowest absolute values, but all showed a strong correlation (r = 0.88–1, Supplementary Materials Figures S5.1 and S5.2). All the differentiation metrics also supported lower absolute value with a clear decrease in the variability among loci for the pairwise differentiation between Sandur vs. Svartárvatn (two northern populations), and between Sandur and Klapparós (Figures S5.1 and S5.2).

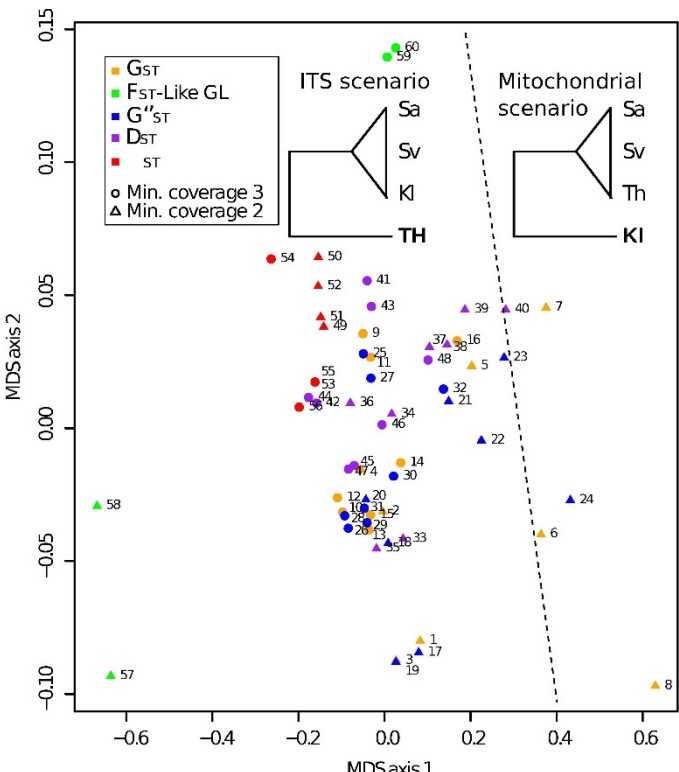

**Figure 3.** Multidimensional scale plot of the correlations between five differentiation indices based on the divergence among the four sampling locations of *Crangonyx islandicus* and various RADseq datasets. Numbers of RADseq datasets are reported in Table S5.2. Trees present the two alternative scenarios based on the mtDNA and ITS genetic variation. Kl: Klapparós, Sa: Sandur, Sv: Lake Svartárvatn, Th: Lake Thingvallavatn.

Two of the three loci detected under putative selection, the mtDNA NADH dehydrogenase subunit 5 and histone 3 loci, were successfully amplified and sequenced from 73 and 79 individuals, respectively, from 14 sites representing the five mitochondrial clades within Iceland (see Supplementary Materials Table S4.1). The NADH sequences of 620 bp showed the same phylogeographic pattern as found earlier by Kornobis et al. [8], based on COI and 16S, with distinct monophyletic mtDNA lineages in the southwest (A, $n = 33$), south (B/C, $n = 7$) and southeast (D, $n = 25$), which were more similar to each other than to the samples from the north (E, $n = 8$, Supplementary Materials Figure S5.3). Unfortunately, no NADH sequences were obtained from the population from Klapparós in the northeast (F, Table S4.1). The variation in histone 3 was confined to a single SNP (A/G), which was variable in the RADseq data with fixed differences between southern and northern Iceland, with the G allele present only in the population from Lake Thingvallavatn, while the A allele was fixed in the populations from Svartárvatn, Sandur and Klapparós (Figure S5.4). Partial histone H3 sequences (314 bp) from 79 individuals belonging to mtDNA lineages A–F showed similar results with only the G allele in the southwest (clade A, A′) and the south (clade B), a large majority of G (44 G and 4 A) in the southeast (clade D) and a huge majority of A (56 A and 4 G) in the north (clade E, F).

### 3.2.3. Congruence Tests among Loci

Considering all datasets, $57.4 \pm 5.0\%$ of the loci on average supported the ITS scenario rather than the mitochondrial scenario for the $G_{ST}$, $G''_{ST}$ and $D_{ST}$. This proportion increased to $74.9 \pm 4.0\%$ on average when considering the $\Phi_{ST}$. However, no significant difference was found for the extent of differentiation between the two scenarios for any of the datasets and differentiation metrics (Wilcoxon test $p > 0.1$).

### 3.2.4. Population Structure and Relationship among Groups Using SNP and Haplotype Frequencies

The K-means clustering approach, based on the RADseq data, retained two clusters systematically, except for the haplotype datasets using a minimal coverage of two and after removing the loci in low frequency and under selection. For these two exceptions, the BIC inferred by sequential K-means never reached clear minima and supported the maximum number of clusters allowed (i.e., 10). On average, the first two axes of the DAPC retained 18.3 ± 7.2% of the total variance. The clustering segregated the individuals into two groups, Thingvallavatn (southwestern Iceland) vs. the three populations from northern Iceland. The assignment plot shows that the level of admixture is rather weak between these two groups, and three to five individuals present in Thingvallavatn can be considered admixed depending on the dataset (Figure 4a–c). Regardless of the dataset, considering sampling location as the a priori number of clusters showed higher levels of admixture, especially among the three northern samples (Figure 4b–d).

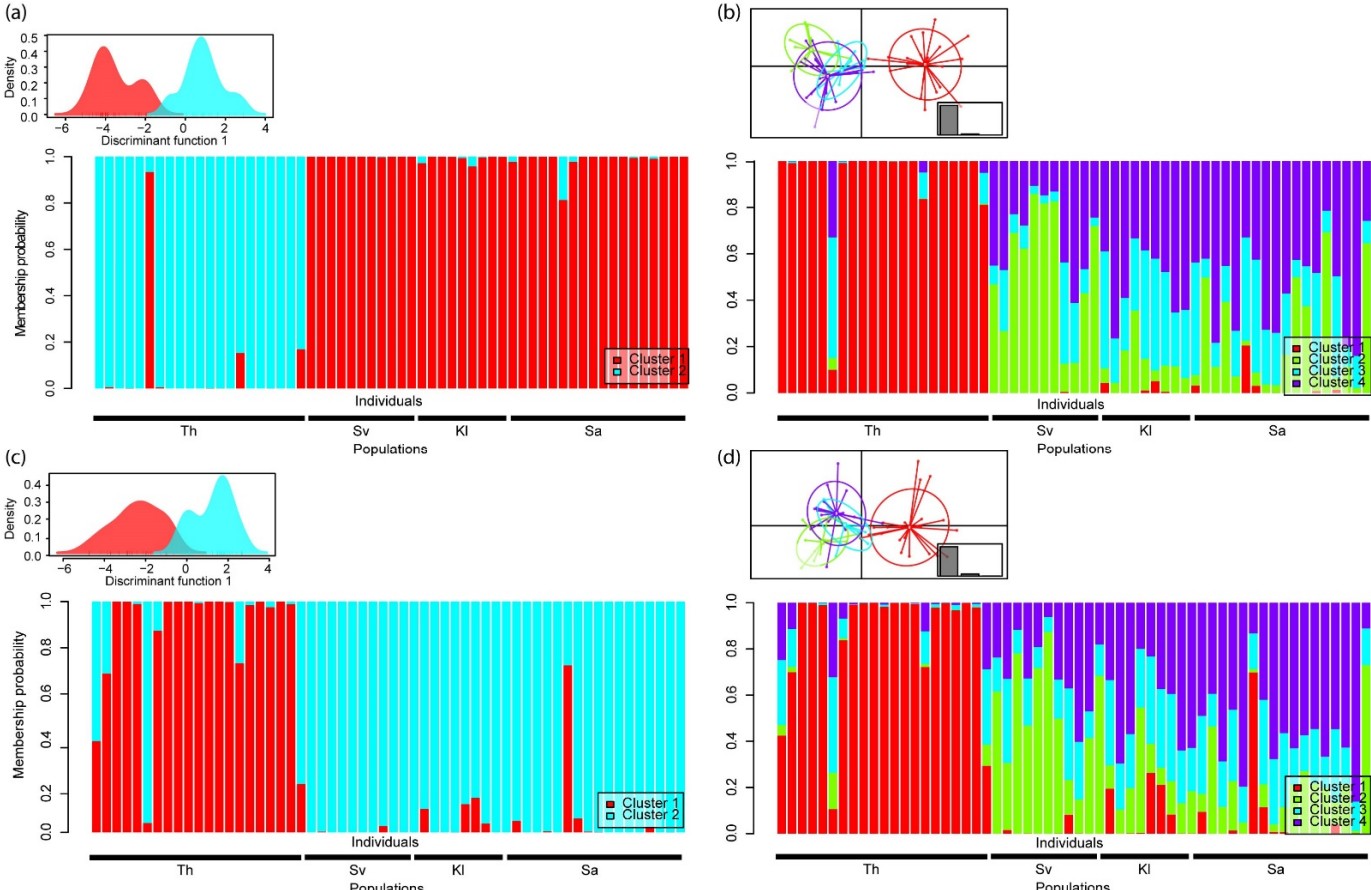

**Figure 4.** Discriminant analysis of principal components (DAPC) of RADseq data of four sampling locations of *Crangonyx islandicus*. The analysis is based on haplotypes, indels as missing data and with a minimum coverage of three (C3Hmiss dataset). The plots in (**a**,**b**) display the results of the C3Hmiss dataset before removing the loci in low frequency (*n* = 103) and after (*n* = 71): (**c**,**d**). The plots in (**a**,**c**) display the results of the optimal number of clusters inferred by a sequential K-means approach according to BIC criteria. The plots in (**b**,**d**) display the results of four sampling locations used as a priori number of clusters. The scatter plot of the DAPC is displayed as small plots and presents the main discriminants axis (1 axis for (**a**,**c**), and 2 axes for (**b**,**d**)), the ellipses represent the inertia of the clusters and the dots represent individuals. The barplots show the assignment probability of each individual to the different clusters. Th: Lake Thingvallavatn, Sv: Lake Svartárvatn, Kl: Klapparós, Sa: Sandur.

The Bayes Factor Delineation (BFD*) approach based on the bi-allelic datasets with the five most informative individuals per location favored the model of four distinct populations (M1), showing a strong support against M2 (three populations with the mitochondrial scenario) and a very strong support against M3 (two populations with the ITS scenario) for both a coverage of two and three (Table 2). However, the topology of the most supported four-population species tree supported clearly the ITS scenario, with the northern populations more closely related together than with the Thingvallavatn population (Figure 5 and Figure S5.5). The posterior probability of the node supporting the northern clade is high (C3SNPr dataset $p = 0.97$, $n = 88$; C2SNPr dataset $p = 1$, $n = 223$), but the node support of the clade E (from Sandur vs. Svartárvatn) is very weak (C3SNPr dataset $p = 0.77$, $n = 88$; C2SNPr dataset $p = 0.59$, $n = 223$). However, the confidence intervals of the date divergence between Thingvallavatn and the three other northern populations overlap, showing a large variance in the coalescence process (Figure 5a). The posterior distributions of $\theta$ of the populations overlapped as well and showed similar $\theta_{median}$ between 0.10 and 0.13 (Thingvallavatn $\theta_{median} = 0.13$, 95% HPD = 0.08–0.19; Klapparós $\theta_{median} = 0.10$, 95% HPD = 0.05–0.16; Sandur $\theta_{median} = 0.12$, 95% HPD = 0.07–0.19; Svartárvatn $\theta_{median} = 0.11$, 95% HPD = 0.06–0.17, see Figure 5b).

**Table 2.** Bayes Factors (BF) of the Bayes Factor Delineation (BFD*) approach [75,76] testing the support of three models of population structure of *Crangonyx islandicus* using RADseq data on unlinked bi-allelic SNPs after removing low-frequency SNPs considered as potential paralogs or putatively under selection. We used the scale of Kass and Raftery [77] to evaluate the support of the best model. The support for model 1 is strong for BF > 2, very strong for BF > 6 and decisive for BF > 10, with BF computed using the formula BF = 2*($ML_{model1} - ML_{model2}$), with ML for marginal likelihood expressed in log scale. SD: Standard deviation.

| Dataset | Minimum Coverage | Number of Unlinked Bi-Allelic SNPs | Models | Averaged Marginal Likelihood among Runs (SD) | Test | BF |
|---|---|---|---|---|---|---|
| C3SNPr | 3 | 88 | 1 (4 populations) | −549.91 (0.09) | M1 vs. M2 | 9.84 |
| | | | 2 (3 populations mitochondrial scenario) | −554.83 (0.20) | M2 vs. M3 | 71.54 |
| | | | 3 (2 populations ITS scenario) | −590.61 (0.16) | M1 vs. M3 | 81.38 |
| C2SNPr | 2 | 223 | 1 (4 populations) | −1245.15 (0.14) | M1 vs. M2 | 179.02 |
| | | | 2 (3 populations mitochondrial scenario) | −1334.59 (0.09) | M2 vs. M3 | 332.53 |
| | | | 3 (2 populations ITS scenario) | −1500.91 (0.02) | M1 vs. M3 | 511.55 |

### 3.2.5. Introgression Tests

The introgression tests based on the D statistic showed a comparable amount of introgression among the northern populations of C. islandicus (from Sandur and Svartárvatn) with either the southern (from Thingvallavatn) or the northeastern population (from Klapparós). A test based on the conservative dataset, with a minimal coverage of three (including 21 to 45 loci) and the mitochondrial scenario with Klapparós population as an outgroup, supported an introgression between Thingvallavatn and Sandur populations, in the genetic combinations of 34 individuals for [Sv,Sa,Th][Kl] but 28 individuals when the structure of the ingroup in the test was changed [Sa,Sv,Th][Kl] (Table 3). Only one combination supported an introgression between Thingvallavatn and Svartárvatn populations. The tests based on the ITS scenario (i.e., with the Thingvallavatn population as an outgroup) showed a similar support of introgression among the northern populations with 28 or 25 combinations of individuals, indicating an introgression between Svartárvatn and Klapparós populations, but only one combination between Sandur and Klapparós populations (Table 3). However, when considering additional loci with a minimal coverage of two (between 62 and 128 loci were used for comparisons), a small number of combinations of individuals supported an introgression between Thingvallavatn and Sandur (five), Thing-

vallavatn and Svartárvatn (four or five), Klapparós and Svartárvatn (five or seven) and Klapparós and Sandur populations (three or four, Table 3).

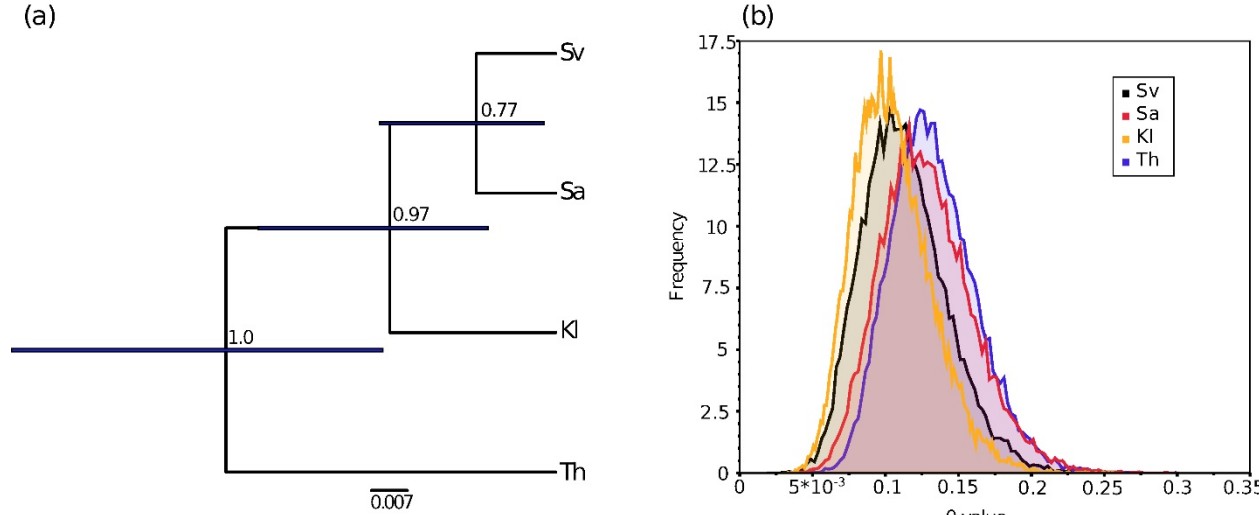

**Figure 5.** Divergence of RADseq genotypes of *Crangonyx islandicus* from the four sampling locations. (**a**) Maximum clade credibility tree (MCCT) and (**b**) posterior distribution of theta ($\theta = Ne*\mu$) parameters inferred by SNAPP [73] according to the Bayes Factor Delineation's (BFD* [75,76]) best scenario for the four populations using 88 unlinked bi-allelic SNPs dataset (C3SNPr). Node number on the MCCT displays the posterior probability of the node, and the dark blue node bar indicates the 95% HPD interval of the node age. Th: Lake Thingvallavatn, Sv: Lake Svartárvatn, Kl: Klapparós, Sa: Sandur.

**Table 3.** Results of the four-taxon D statistic testing for introgression among sampling locations of *Crangonyx islandicus* using RADseq data. P1, P2 and P3 are the populations under investigation for introgression and O is the outgroup. Range Z is the range of the Z value, upper values deviated significantly from 0 ($p < 0.01$). NSign/Ntot for positive D reports the number of significant replicates (combination of individuals) over the total number of tests for positive D statistic indicating an introgression between P2 and P3. NSign/Ntot for negative D indicates an introgression between P1 and P3. Range NbLoci is the range of loci used among replicates to test the D statistic. Th: Lake Thingvallavatn, Sv: Lake Svartárvatn, Kl: Klapparós, Sa: Sandur. Min. cov. presents the minimum coverage of datasets 3 (Cov3Hmiss_cl) and 2 (Cov2Hmiss_cl).

| Min Cov. | Test | P1 | P2 | P3 | O | Range Z | NSign/Ntot for Positive D | NSign/Ntot for Negative D | Range NbLoci |
|---|---|---|---|---|---|---|---|---|---|
| 3 | Mito.1 | Sv | Sa | Th | Kl | 0–10.1 | 28/125 | 0/125 | 33–45 |
| 3 | Mito.2 | Sa | Sv | Th | Kl | 0–10.1 | 1/125 | 34/125 | 33–45 |
| 3 | ITS.1 | Sv | Sa | Kl | Th | 0–10.1 | 28/125 | 0/125 | 21–43 |
| 3 | ITS.2 | Sa | Sv | Kl | Th | 0–8.2 | 25/125 | 1/125 | 21–43 |
| 2 | Mito.1 | Sv | Sa | Th | Kl | 0–7.7 | 5/125 | 4/125 | 101–128 |
| 2 | Mito.2 | Sa | Sv | Th | Kl | 0–8.2 | 5/125 | 5/125 | 101–128 |
| 2 | ITS.1 | Sv | Sa | Kl | Th | 0–6.4 | 7/125 | 3/125 | 62–128 |
| 2 | ITS.2 | Sa | Sv | Kl | Th | 0–6.0 | 5/125 | 4/125 | 62–128 |

## 4. Discussion

The current study showed the robustness of the mito-nuclear discordance of the phylogeographic history of *C. islandicus*, as newly sequenced individuals from previously

described populations for mitochondrial DNA confirmed the previous mitogenomic findings by Kornobis et al. [8], while a multitude of nuclear genomic markers sequenced through ddRADseq provided further support to the ITS scenario [9]. A structured coalescent analysis using the mitochondrial 16S and CO1 sequences from the same four populations assessed with ddRADseq data confirmed the previous mitochondrial pattern and supports a clear pre-Ice Age divergence between the northeastern Icelandic (from Klapparós) population and the northern and southwestern populations. Despite the use of multiple ddRADseq analytical approaches designed to overcome potential misinterpretations, including different data types (SNPs, haplotypes, DNA sequences), different dataset configurations (brute vs. clean), base call uncertainty and various metrics, most approaches (differentiation indices, DAPC, BFD*) were congruent and supported the ITS scenario, with the main divergence occurring between the northern vs. southern populations. We showed that a higher proportion of the RADseq loci followed the ITS scenario compared to the mitochondrial scenario, which was still strongly recovered by a small proportion of RADseq loci. The admixture revealed by DAPC was higher among northern populations than between southern and northern populations (admixture was found for several individuals though), which globally provides support to the ITS scenario. A specific approach used to disentangle the contribution to ILS and introgression/ancestral hybridization related to these discordant genealogies showed a comparable introgression signal between populations in northern (from Sandur and Svartárvatn) and northeastern Iceland (from Klapparós) and between the population in the south (from Thingvallavatn) and the ones from the north. Overall, despite the ITS scenario receiving greater support from the RADseq data, our results suggest a complex scenario involving not only incomplete lineage sorting but also possibly introgression led by male-biased dispersal among northern locations (but see the discussion below) or mitochondrial capture and then the accelerated isolation of the mitochondrial DNA among the northeastern, southern and southwestern populations possibly strengthened by selection, a scenario that would require additional research to be fully tested.

### 4.1. Methodological Uncertainties with RADseq Data

As a genome reduction approach, RADseq analysis assesses variation from small genomic fragments sampled from throughout the genome without any a priori information about their mode of inheritance, location, substitution rate or recombination rate. Overall, the RADseq loci may present an important variability among all these characteristics that may affect commonly used summary statistics in population genetics when population differentiations and other population parameters are estimated. Our results reveal no systematic bias for a particular differentiation metric; however, they confirm the potential higher sensitivity of $G_{ST}$ to loci with high genetic diversity and may thus affect the estimate of population structure as the proportion of such markers increases (cf. [30]). Only five out of sixty statistics calculated for the subdivision of the populations supported the mitochondrial scenario; the other supported the ITS scenario. Of the five statistics supporting the mtDNA, three were based on GST, which does not take into account the dependency of the statistic on the overall variation as $G''_{ST}$ and $D_{ST}$ do, which may thus give a biased overall estimate. Furthermore, these five methods are based on the lower coverage datasets (minimal coverage two), which included a higher number of loci but at the price of higher base call uncertainty, and four of them included indels as the 5th state. Higher coverage and omitting indels is a more conservative approach, as increasing coverage offers a more robust estimate of the DNA sequence polymorphisms and indels can lead to problems in alignment and may reflect errors in sequencing [80]. Thus, those five approaches (dataset and metrics) supporting the mtDNA-scenario may be clearly suboptimal to assessing the population structure. Specific methods accounting for uncertainty related to low-coverage data (regardless of the minimal coverage) based on a maximum likelihood base call estimate (ANGSD) confirmed the ITS population structure scenario, also inferred by most classic approaches not accounting for such a bias.

Methods relying on genotype likelihood instead of SNP calling to infer population statistics offer great promise to considerably decrease the bias caused by base call uncertainty, even for very low-coverage datasets (i.e., x2) but remain limited to estimating basic summary statistics [27,39,41,43]. However, our results suggested that the potential noise brought by low-coverage data may not be strong enough to blur the main phylogeographic signal. These results are in agreement with the findings of several studies [36,37], reporting that when facing the trade-off of sampling more individuals per population or increasing the per sample depth coverage to infer population parameters, the former offers the best strategies because each sequence read from a new individual brings more information than additional reads (considering that the reads with a low coverage are discarded for the analyses) already present in the pool [36]. In addition, the trade-off between the number of individuals per population vs. the depth coverage of the DNA loci can be further tangled by the genome size of the organism [81]. Indeed, organisms with large genomes may increase the number of restriction sites and thus DNA loci, which overall limit the number of read copies per loci.

Although the genome size of *C. islandicus* is currently unknown, we suspect it could be large, which could partly explain the low coverage of our dataset. Indeed, crustacean and amphipod species in particular are prone to gigantism in their genome size, especially species supposedly with low metabolic rates [82] present in cold water environments such as the Arctic [83,84] and deep waters from Lake Baikal [85], which share many similarities with the groundwater habitats of *C. islandicus* [8,45,46]. Moreover, groundwater species tend to have larger genome sizes than their surface sister species due to the reduction in effective population size [86].

When the information in a DNA sequence was reduced to a haplotype or a single SNP per locus, the results did not show any systematic bias toward a particular scenario. However, the use of substitution models including the DNA sequence of the fragments such as in the $\Phi_{ST}$-index, showed consistently robust patterns regardless of coverage (2 or 3), or when removing loci in low frequency, potential paralogs or loci under selection. Further, our results confirm the increase in power to detect candidate loci showing the signature of selection when using haplotype data inferred from the full sequence of the RADseq loci in comparison with the use of a single SNP per locus [87,88]. Considering indels as a fifth state when building the haplotypes might provide additional power to further detect loci under selection, as suggested by our results.

*4.2. The Phylogeographic Pattern of Crangonyx islandicus*

We have shown here that different analyses based on the extended mtDNA dataset and a larger nuclear dataset (ddRADseq) than in [8,9] confirmed the different mitochondrial and nuclear phylogeographic patterns in *C. islandicus*. Mito-nuclear discordance among structured populations or closely related taxa has been detected in several taxonomic groups [1,6,7], including a groundwater species complex belonging to genus *Crangonyx Bate*, 1859 [89]. Disentangling the phylogeographic scenario and its causes remains extremely difficult, as many mechanisms can lead to the same patterns. However, from the previous and present results it is clear that this groundwater species diverged within Iceland through the Pleistocene, when Iceland was repeatedly covered by glaciers [46,90]. The main split observed for most genomic markers, the ITS scenario, between southern and northern Iceland, reflects different watersheds separated by the interior highlands of the country, whereas the earliest split in the mtDNA was between the northern and northeastern areas followed by the subsequent structuring of the southeast, south and southwest of the country from an ancestor of the northern population. We propose below tentative and non-mutually exclusive phylogeographic scenarios implying several mechanisms commonly proposed in the literature to explain such complex mito-nuclear discordant patterns [6,7] with some explanation about their likelihood.

A first scenario, based only on stochastic forces driven by a strong incomplete lineage sorting between the mitochondrial and most nuclear markers, sounds appealing at first

but it seems relatively unlikely. Indeed, mito-nuclear discrepancies may be caused by the lower population effective size (Ne) of the mitogenome ($1/4$ of the nuclear), which makes it more likely to be quickly reciprocally monophyletic (shorter coalescence times due to faster loss of ancestral polymorphisms) [6]. Mitochondrial DNA has thus been referred to as a 'leading' indicator, rather than 'lagging' as for nuclear markers, which more often show paraphyletic or polyphyletic relationships because nuclear markers take more time to coalesce [1]. In addition to a less effective population size (Ne) in mtDNA, background selection or hitch-hiking might further reduce Ne or the variation within populations, e.g. [13], and thus to increased population differentiation, as estimated with $F_{ST}$, which is not yet apparent with nuclear DNA, especially if the ancestral population had a large Ne. The decoupling of the mitochondrial vs. nuclear substitution rates has been reported in many phyla [91], but a recent study showed a lower nuclear substitution rate for subterranean organisms than for surface organisms due to a lower metabolic rate and reduced genome replication rate, while the substitution rate of the mitochondrial genome remains unaffected [92]. This would further increase the differences in substitution rates between the mitochondrial and nuclear genomes for subterranean taxa. The populations from the north (Sandur and Svartárvatn) and northeast (Klapparós), currently located on each side of the volcanic active zone characterized by a high geothermal area and volcanic activity north of Lake Mývatn, may share a common origin further south in the interior of the country. The divergence between these northern and northeastern populations might be too recent for the nuclear DNA that has not yet achieved a complete lineage sorting (i.e., ILS) and showed strong admixture, while the mitochondrial genome reaches reciprocal monophyly much more quickly.

Even though ILS may have played a role in the mito-nuclear discordance, we propose a second scenario involving introgression among populations previously isolated due to the opening of dispersal corridors that may not be conducive anymore. Such a scenario might be especially prevalent in the interior of the country, where geographical distances are shorter and individuals from populations characterized by different mtDNA lineages are more likely to meet. Furthermore, clear phylogeographic mito-nuclear discordant patterns usually rule out the ILS scenario [6]. Indeed, it is difficult to explain the strong and discordant phylogeographic structure among the populations, especially the main and deep split of the nuclear DNA between the northern and southern populations (i.e., Thingvallavatn) without involving introgression.

The DAPC admixture plots and introgression test results supported a certain amount of ancestral hybridization, especially between the populations in the north (from Sandur and Svartárvatn) and the northeast (from Klapparós), but also between the populations in the south (Thingvallavatn) and the north, which implies the presence of secondary contacts [13,26,79] and thus the opening of dispersal corridors. The two events may though have led to different incongruences between the mtDNA and the nuclear genomes. The discordant pattern between the northern and northeastern populations suggests that introgression occurred mostly on the nuclear genome and not on the mitochondrial genome that shows a clear phylogeographic pattern of reciprocal monophyly. This might have resulted from male-biased dispersal, but no males have been identified within the species [44] and the sex differences may be subtle. The mtDNA from the Klapparós population might also have been captured and replaced by a distinct unsampled ancestral population.

In the study by Kornobis et al. [8], the largest variation was observed at higher altitudes, suggesting these sites were the sources for populations at lower altitudes (<100 m), which were below sea level about 10,000 years ago (see references in [8]). Both mtDNA and RADseq data showed a close genetic proximity among the two northern populations from Sandur in Adaldal and Lake Svartárvatn, which may be explained by the recent colonization of Sandur, which is a low altitude area that was submerged by sea at the end of the last glacial period of the Ice Age about 10 kyr ago. Sandur is likely to have been colonized from the areas around lakes (Svartárvatn and/or Mývatn) further south. The sites are linked today by rivers and major lava fields, which ran to the north along the valleys Bárdardalur

(9500 and 9000 years ago) and Laxárdalur (5000 and 2200 years ago) [93]. It is though noteworthy that samples from the interior of Iceland, where the different evolutionary lineages are separated by shorter distances and where crossing may have occurred, are scarce. The clustering of the H3 gene from the small sample from the highlands, Kjölur (mtDNA C), with southeast Iceland (mtDNA D) indicates that reproduction between the two lineages may have happened there. A similar scenario might have happened south of Lake Mývatn, where the ancestral admixture of samples now inhabiting the sampled sites in northern and northeastern Iceland might have occurred. Although the divergence in mtDNA between samples from different locations follows well the geographic distances and geological features, there are some exceptions that point to the fact that dispersal can occur via rivers on the surface or along lava fields, whose topology and thus migratory routes are likely to have changed considerably over time during the different glacial and interglacial periods of Iceland, subsidence, post-glacial rebound and volcanic activities. One example is the distribution of the BC mtDNA haplotypes, which are found in separate regions of the volcanic zones in southern Iceland [8].

Ancient introgression followed by isolation and the differentiation of the mtDNA lineages among northern populations offers an appealing scenario for the split among northern populations but it does not yet explain why nuclear DNA showed the main and deep split between the northern and southern populations (i.e., from Lake Thingvallavatn). Interestingly, both the mtDNA and RADseq data showed that the southwestern population from Thingvallavatn harbored the highest theta (a proxy of Ne assuming a constant mutation rate), especially for the mtDNA, which was almost four and seven times higher than the theta of the populations from the northeast (Klapparós) and the north (Sandur and Svartárvatn), respectively. Populations with the highest Ne or genetic diversity usually suggest a long presence of a large population (center of diversification) or they can also correspond to a secondary contact zone among previously isolated populations [94,95]. Southwestern Iceland (i.e., from Thingvallavatn) represents a large area of potential habitats for *C. islandicus,* which may have maintained several populations through time [8]. The genetic diversity of the Lake Thingvallavatn population, and thus its Ne, may have been fed by regular connections and secondary contacts with peripheral populations belonging to the mitochondrial clades A, A' and B [8]. Those peripheral populations might be regularly connected and isolated depending on the opening/closing of fissures due to geothermal and tectonic activities in the regions [8]. The long presence of the southwestern population, and possibly northeastern populations, could explain the deep divergence in the nuclear genome. Moreover, the closer similarity of the mtDNA clades in northern and southern populations compared to the one in the northeast could also indicate an ancestral hybridization, where mtDNA was more likely to introgress or be captured by one lineage [6] but has then diverged more rapidly than the nuclear genome due to smaller Ne, which could be strengthened by selection [7,96].

## 5. Conclusions

Well-supported inferences about the phylogeographic patterns of populations rely on good knowledge about the genomic and geographic variation within species. Here we show that the genome-wide markers obtained with RADseq analysis provide greater support to previous results on the nuclear ITS region [9] rather than the results from the mitochondrial analyses, albeit a large variation among loci. The main genetic divergence within *C. islandicus* follows the split between northern and southern Iceland, but the analyses indicate the evidence of introgression between the main evolutionary lineages within the country. The analysis of whole genome variation could further give insights into the evolutionary histories of these populations. Although RADseq analyses provide information from a large number of markers, information on their location within the genome could allow further inferences, which could take into account the possible impacts of varying effective size and selection due to variation in ploidy, structural variation or linkage. Furthermore, increased geographic sampling, especially from the interior

of Iceland, may reveal more mixed populations and thus insight into the history of the populations that have been studied and are mostly found in the lowland areas of the country. Overall, our study paved the way to combine large numbers of both nuclear and mitochondrial markers to disentangle species' evolutionary history and revisit previous mitochondrial phylogeographic scenarios, especially among closely related species, as a more complex picture may emerge.

**Supplementary Materials:** The following supporting information can be downloaded at: https://www.mdpi.com/article/10.3390/d15010088/s1, Supplementary Materials S1—Sampling information; Table S1.1: Sampling information on 59 individuals of *Crangonyx islandicus* obtained for RADseq analyses and to complement mitochondrial analyses; Table S1.2: Information on 37 individuals of *Crangonyx islandicus* extracted from Kornobis et al. [1] used to complete the MultiTypeTree structured coalescent analysis. Supplementary Materials S2—Adapter and barcode sequences; Supplementary Materials S3—Protocol of the SYBR gold fluorometric assay for dsDNA quantitation; Supplementary Materials S4—Additional analysis of the NADH and H3 loci showed indication of selection in RADseq analysis [97–99]. Table S4.1: Samples obtained for sanger sequencing for NADH dehydrogenase subunit 5 and the histone 3 (H3), which showed indication of selection in RADseq analysis. Figure S4.1: Sampling sites of *Crangonyx islandicus* in Iceland. Further description of the sites is given in Table S4.1. Dark grey zone presents the volcanic zone while glaciers are in light grey. Table S4.2: Primers used for amplification of mtDNA NADH dehydrogenase subunit 5 and the histone exon 3 (H3). The sequence length of NADH5 was 760 bp and of H3 was 314 bp. Supplementary Materials S5—Additional results. Table S5.1: Blastn results of the RADseq loci detected under selection by Bayescan. Table S5.2: Overall pairwise differentiation measures among populations of *Crangonyx islandicus* inferred using RADseq data allowing the correspondence between the labels in Figure 3 and the type of data used. Bold values show significant *p*-value at $\alpha = 0.05$ pairwise $G_{ST}$, $G''_{ST}$, $D_{ST}$ and $\Phi_{ST}$ among populations using a combined Fisher's *p*-value among loci corrected for a multiple testing using Holm's formula. Clean_Dataset: using a brute (br) dataset coming from pyRAD or a cleaned (cl) dataset removing loci with a Minimum Allele Frequency (MAF) below 5%, with an observed heterozygosity above 0.5 and detected under selection by Bayescan. Th: Lake Thingvallavatn, Sv: Lake Svartárvatn, Kl: Klapparós, Sa: Sandur. Figure S5.1: Global and pairwise differentiation values of each RADseq locus among the four populations of *Crangonyx islandicus*. $D_{ST}$, $G_{ST}$ and $G''_{ST}$ are computed using haplotype information of a dataset considering indels as missing data and a minimal coverage of the RADSeq loci of 3 (C3Hmiss dataset). The $\Phi_{ST}$ were computed using the sequences of the same dataset. The dataset C3Hmiss before (a) (*n* = 103), and after (b) removing the loci in low frequency, considered as potential paralogs or under selection (*n* = 71). Th: Lake Thingvallavatn, Sv: Lake Svartárvatn, Kl: Klapparós, Sa: Sandur. The black bands display the median distance, the box represents the interquartile distance, the whiskers are long up to 1.5 the interquartile distance while the black dots represent the outliers. Figure S5.2: Global and pairwise differentiation values of each RADseq locus among the four populations of *Crangonyx islandicus*. $D_{ST}$, $G_{ST}$ and $G''_{ST}$ are computed using haplotype information of a dataset considering indels as missing data and a minimal coverage of the RADSeq loci of 2 (Cov2Hmiss dataset). The $\Phi_{ST}$ were computed using the sequences of the same dataset. The dataset Cov2Hmiss before (a) (*n* = 295), and after (b) removing the loci in low frequency, considered as potential paralogs or under selection (*n* = 202). Th: Lake Thingvallavatn, Sv: Lake Svartárvatn, Kl: Klapparós, Sa: Sandur. The black bands display the median distance, the box represents the interquartile distance, the whiskers are long up to 1.5 interquartile distance while the black dots represent the outliers. Figure S5.3: Phylogenetic tree of 73 individuals of *Crangonyx islandicus* based on 620 bp of the partial mtDNA NADH dehydrogenase subunit 5 gene detected as putatively under selection by Bayescan. This maximum likelihood tree was built with Phyml [98] using a GTR evolution model. Tips represent the different haplotypes and tip labels show the label of the site location as described in Table S4.1 and the number of individuals (i.e., frequencies) belonging to a given haplotype is provided within brackets. The branch support (aLRT) displayed at nodes was obtained with the difference in likelihoods with and without the respective branch. Geographic origin is presented with vertical bars using the following letters: SV: southwestern Iceland, S: southern Iceland, SA: southeastern Iceland and N: northern Iceland. Site G is from Kjölur, central Iceland (see Figure S4.1). Figure S5.4: Phylogenetic tree of 79 individuals of *Crangonyx islandicus* based on 374 bp of partial histone 3 (H3) gene detected as putatively under selection

by Bayescan. This maximum likelihood tree was built with Phyml [98] using a GTR evolution model. Tips represent the different haplotypes and tip labels show the label of the site location as described in Table S4.1 and the number of individuals (i.e., frequencies) belonging to a given haplotype is provided within brackets. The branch support (aLRT) displayed at nodes was obtained with the difference in likelihoods with and without the respective branch. Geographic origin is presented with vertical bars using the following letters: SV: southwestern Iceland, S: southern Iceland, SA: southeastern Iceland and N: northern Iceland. Figure S5.5: (**a**) Maximum clade credibility tree (MCCT) and (**b**) posterior distribution of theta ($\theta$ = Ne*μ) parameters inferred by SNAPP [74] according to the Bayes Factor Delineation's (BFD* [76]) best scenario for the four populations of *Crangonyx islandicus* using 223 unlinked bi-allelic SNP dataset (C2SNPr). Node number on the MCCT displays the posterior probability of the node and the dark blue node bar presents the 95% HPD interval of the node age.

**Author Contributions:** D.E. and S.P. performed the conceptualization and designed the study, B.K.K., S.P. and D.E. performed the sampling and the morphological identification of the individuals, K.M.W., D.E. and B.M. performed the laboratory work, D.E., K.M.W. and S.P. defined the methodology, D.E. and S.P. analyzed the data, D.E. performed the visualization, D.E. and S.P. led the writing, S.P. secured the funding. All authors have read and agreed to the published version of the manuscript.

**Funding:** The study was supported by Research Grant No. 130244-05, funded by the Icelandic Research Fund.

**Data Availability Statement:** Publicly available datasets were analyzed in this study. The data and parameter files, as well as R scripts and bash codes used to perform the analyses of the mitochondrial and the RADseq data, are available at: the Zenodo Digital Repository https://doi.org/10.5281/zenodo.7506345. The RADseq data are available in NCBI with the BioProject number PRJNA917071. The new CO1, 16S, NADH5 and H3 DNA sequences have been deposited in GenBank and accession numbers are available in Supplementary Materials Tables S1.1 and S4.1.

**Acknowledgments:** We would like to thank Christophe J. Douady, Tristan Lefébure and Agnes-Katharina Kreiling for their help in the field.

**Conflicts of Interest:** The authors declare no conflict of interest.

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
