# Peer review of "Contrasting Phylogeographic Patterns of Mitochondrial and Genome-Wide Variation in the Groundwater Amphipod Crangonyx islandicus That Survived the Ice Age in Iceland"

_diversity, doi:10.3390/d15010088_

Round 1
Reviewer 1 Report
I have enjoyed reading the manuscript, in which the authors explored the mito-nuclear discordance patterns in an amphipod species Crangonyx islandicus. The historical distribution of this object of the study is extremely interesting, as it is probably the only metazoan species that survived the Ice Age under glaciers. The authors used ddRADseq to test the contrasting hypotheses that came from previous analysis of several mitochondrial and nuclear markers.
=== General comments
In general, the manuscript is written clearly, and the results are convincing. However, I feel that it could benefit from some clarifications.
L167: “the CO1 and 16S genes and sequences added from 37 individuals previously sampled in these 4 locations [8].”: I found this data reference quite confusing and am afraid that it renders the analysis not exactly reproducible. On the one hand, the Kornobis et al. 2010 paper mentions more than 37 sequenced individuals even for the discussed A, A’, E and F haplogroups, and at the same time I could found 35 COI and 17 16S sequences stored in NCBI for all haplogroups. Where did the additional sequences come from? I would strongly recommend clearing up this issue, providing the IDs of the particular sequences used from Kornobis et al. 2010 in a supplementary file, for example, and deposit missing sequences to NCBI if necessary.
There are several other issues related to the NCBI deposition. First, I would highly encourage the authors to mention the sampling place for each sequence to avoid confusion in the future. Second, I recommend to deposit the raw sequencing data for each library as well for the purposes of reproducibility.
Moreover, the whole analysis heavily relies on in-house scripts and fine tuning of programs. Even though I appreciate that the authors mention all program versions, I would highly recommend adding custom scripts and parameter files, such as xml files for Beauti, into a repository such as Github, Zenodo etc. As the authors test, as they write, indeed a new generation tool and explore methodology-related questions, a collection of reproducible scripts for data analysis would greatly benefit the other researchers.
That being said, I would like to commend the authors for including detailed protocols in the supplementary materials.
=== More specific comments (minor issues)
Title: “mitochondrial and genomic variation”. I feel that this wording is unclear: mitochondrial genome is still part of a genome. However, I can understand why avoiding the mitochondrial vs nuclear distinction, as the ddRADseq data contained some mitochondrial markers as well.
Maybe “mitochonrial and genome-wide variation” would work better?
The description of sampling is slightly confusing. Compare:
L244-245: “amplifying and sequencing specimens from 73 individuals from 14 sites”
L498: “73 and 79 individuals coming from 15 sites”
So, were 73 or 79 individuals sampled?
L44-46: the referenced examples of mito-nuclear discordance come from general review on the subject or objects quite different from the object of this study with differing dispersal strategies. Does it mean that this event has not been explored in groundwater and other amphipods / other groundwater crustaceans so far? If there is data, it would be great to add more information on the existing data and the prevalence of mito-nuclear discordance in related taxa.
Methods: shouldn’t 2.3.1 and 2.3.2 be in the opposite order, as they are in the Results section?
Fig. 1a caption “The volcanic active zone is marked with curved dotted lines”: as for me, these lines are quite hard to see. I only realized where they were after looking at the map in Kornobis et al. 2010. Maybe it’s worth changing the chade of gray (to white?), or the thickness, or the dotting patter to make them more visible.
Fig. 1a: why is latitude marked in ° O (not W)?
L405 perfectly support => perfectly supportED (or supports)?
L416 overlapPED?
Author Response
Reviewer 1
I have enjoyed reading the manuscript, in which the authors explored the mito-nuclear discordance patterns in an amphipod species Crangonyx islandicus. The historical distribution of this object of the study is extremely interesting, as it is probably the only metazoan species that survived the Ice Age under glaciers. The authors used ddRADseq to test the contrasting hypotheses that came from previous analysis of several mitochondrial and nuclear markers.
=== General comments
In general, the manuscript is written clearly, and the results are convincing. However, I feel that it could benefit from some clarifications.
L167: “the CO1 and 16S genes and sequences added from 37 individuals previously sampled in these 4 locations [8].”: I found this data reference quite confusing and am afraid that it renders the analysis not exactly reproducible. On the one hand, the Kornobis et al. 2010 paper mentions more than 37 sequenced individuals even for the discussed A, A’, E and F haplogroups, and at the same time I could found 35 COI and 17 16S sequences stored in NCBI for all haplogroups. Where did the additional sequences come from? I would strongly recommend clearing up this issue, providing the IDs of the particular sequences used from Kornobis et al. 2010 in a supplementary file, for example, and deposit missing sequences to NCBI if necessary.
>>> Our reply 1: To clarify our selection of sites and individuals to document the mitochondrial diversity of the mitochondrial clade AA’, E and F, we have provided an additional table in Supplementary Material, the Table S1.2 (see “SuppMat_Table_S1.1.2_S4.1_S5.1_S5.2.xlsx”) including all the 37 individuals retained from the previous study by Kornobis et al. 2010 with the GenBank accession numbers of their haplogroup.
We have provided additional details in the material methods L.165-169 “To validate the mitochondrial pattern observed for these locations by Kornobis et al. [8], 21 individuals out of the 59 individuals were newly sequenced for the CO1 and 16S genes in this study and these sequences were completed by 37 individuals previously sampled in the immediate surroundings of these 4 locations [8] for a total of 58 individuals used in the mitochondrial analysis (see Supplementary Materials Table S1.2).”
Among all the 97 individuals with both the CO1 and 16S among the 5 mitochondrial clade AA, B, C, D, E and F sequenced in Kornobis et al. 2010, we retained only the individuals within the three clade AA, E and F in this study, for the same sampling sites or the closest sampling sites located within the same hydrographic system as the 59 individuals sampled in 2013 for the RADseq (See Supplementary material Table S.1.1, “SuppMat_Table_S1.1.2_S4.1_S5.1_S5.2.xlsx”). For example, for the mitochondrial clade AA we retained only 24 individuals sampled in two sites in springs from the lake Thingvallavatn (location 2 and 3 on the map presented in Figure 1 from the Kornobis et al. 2010 publication).
<<<
There are several other issues related to the NCBI deposition. First, I would highly encourage the authors to mention the sampling place for each sequence to avoid confusion in the future. Second, I recommend to deposit the raw sequencing data for each library as well for the purposes of reproducibility.
>>> Our reply 2: We agree with this comment, and we have now provided three tables in Supplementary Materials to provide sampling information about 1) the 59 new individuals samples in this study allowing to perform the RADseq analyses and providing 21 CO1 and 21 16S new sequences (see Table S1.1), 2) the 37 individuals sampled previously by Kornobis et al. 2010 and their associated CO1 and 16S mitochondrial DNA sequences, including their accession numbers (see Table S1.2), and 3) the 84 individuals previously sampled by Kornobis et al. 2010 that were used in this study to generate new DNA sequences for two DNA regions, NADH5 (73 individuals were newly sequenced for the NADH5) and Histone 3 (H3N2, 79 individuals were newly sequenced for the H3N2) detected under putative selection by Bayescan based on the RADseq analyses (see Table S4.1).
All the new sequences generated by this study were submitted to NCBI, so far, we received the accession numbers for the new CO1, 16S and NADH5 sequences that were included in Table S1.1 and Table S4.1. We are currently waiting to receive the accession numbers for the other sequences that should come in the following days.
We are also in the process of submitting the raw sequencing data from the library of the three runs of sequencing, which will be deposited as SRA on the NCBI in the coming days.
<<<
Moreover, the whole analysis heavily relies on in-house scripts and fine tuning of programs. Even though I appreciate that the authors mention all program versions, I would highly recommend adding custom scripts and parameter files, such as xml files for Beauti, into a repository such as Github, Zenodo etc. As the authors test, as they write, indeed a new generation tool and explore methodology-related questions, a collection of reproducible scripts for data analysis would greatly benefit the other researchers.
>>> Our Reply 3: We have provided all the parameter files, data files and R and bash scripts to perform the mitochondrial and RADseq analyses. For the RADseq analyses we presented all the codes and parameters and datafiles of the different analyses for the dataset with a minimal coverage of 3 as an example. We have also provided the data files for the minimal coverage of 2 and using the scripts and codes example to run the analyses for a minimal coverage of 3, all the results can be reproduced for the minimal coverage of 2 as well. A ReadMe.txt file in the “DataAvailability” zip archive helps to navigate and understand the meaning and the data content of the different folders and files contained in the archive, to help other researchers to benefit from this work.
As stated in the data availability statement and the cover letter, the “DataAvailability” zip archive file is currently available during the review process at the following address https://drive.google.com/file/d/1udO9SbNiIxyXngrrQUmuBDE4bSsUsVD6/view?usp=share_link and will be deposited in the Zenodo Digital Repository upon acceptance.
We have provided further details in the data availability statement L.887-894 “The data and parameter files as well as R scripts and bash codes used to perform the analyses of the mitochondrial, outlier DNA markers and the RADseq data are currently available at: https://drive.google.com/file/d/1udO9SbNiIxyXngrrQUmuBDE4bSsUsVD6/view?usp=share_link and will be deposited in the Zenodo Digital Repository [ref] upon acceptance of the manuscript.“
<<<
That being said, I would like to commend the authors for including detailed protocols in the supplementary materials.
>>> Our Reply 4: We have clarified and provided additional information for several analyses in Supplementary Materials. In addition we believe that by providing the data and parameters files as well as R scripts, in-house R functions and bash codes in an open access archive (See Our reply 3 about the archive of the data and codes), we greatly improve the reproducibility of our analyses and results and that will help others to harness those recent tools to address their own questions.
<<<
=== More specific comments (minor issues)
Title: “mitochondrial and genomic variation”. I feel that this wording is unclear: mitochondrial genome is still part of a genome. However, I can understand why avoiding the mitochondrial vs nuclear distinction, as the ddRADseq data contained some mitochondrial markers as well.
Maybe “mitochonrial and genome-wide variation” would work better?
>>> Our Reply 5: We agree and we changed the title of the manuscript accordingly.
The description of sampling is slightly confusing. Compare:
L244-245: “amplifying and sequencing specimens from 73 individuals from 14 sites”
L498: “73 and 79 individuals coming from 15 sites”
So, were 73 or 79 individuals sampled?
>>> Our Reply 6: We agree, we made a mistake about the number of sites. In the revised version of the manuscript we have clarified this aspect, L,242-245 “Loci identified to be under selection were further analyzed, by designing primers in sequences under selection and by amplifying and sequencing specimens from up to 79 individuals per loci from 14 sites (see Supplementary Materials S4 for additional details) from the species range [8].” and L.491-494 “Two of the three loci detected under putative selection, the mtDNA NADH dehydrogenase subunit 5 and histone 3 loci were successfully amplified and sequenced from 73 and 79 individuals, respectively, from 14 sites representing the five mitochondrial clades within Iceland (see Supplementary Material Table S4.1).”.
<<<
L44-46: the referenced examples of mito-nuclear discordance come from general review on the subject or objects quite different from the object of this study with differing dispersal strategies. Does it mean that this event has not been explored in groundwater and other amphipods / other groundwater crustaceans so far? If there is data, it would be great to add more information on the existing data and the prevalence of mito-nuclear discordance in related taxa.
>>> Our Reply 7: We agree that this is a general issue for sexually transmitted markers and this discussion remains very marginal for groundwater crustacean species to the best of our knowledge beyond Kornobis et al. 2011 and a recent study by Cannizzaro et al. 2020 on another Crangonyx species. However both studies used at best only two nuclear or mitochondrial markers. We have nevertheless included the recent article by Cannizzaro et al. 2020 in the discussion L.682-684 “Mito-nuclear discordance among structured populations or closely related taxa has been detected in several taxonomic groups [1,6,7] including within a groundwater species complex belonging to Crangonyx genus [89].”
<<<
Methods: shouldn’t 2.3.1 and 2.3.2 be in the opposite order, as they are in the Results section?
>>> Our Reply 8: We prefer to keep the original order, by presenting all the bioinformatic steps related to the preparation of the RADseq dataset files in the 2.3 “Bioinformatic pipeline” paragraph (including the 2.3.1 about “2.3.1 SNPs, haplotypes and sequence datasets base-called with pyRAD.” and the 2.3.2 about “Genotype likelihood datasets with ANGSD”). We think it is clearer to present this before we describe the statistical analyses of both the mitochondrial datasets 2.4 and the RADseq datasets 2.5. On the contrary, we found it easier to present all the results of the mitochondrial analyses first and then the results of the RADseq analyses as we did for the presentation of the mitochondrial and RADseq analyses in the Material and Methods (paragraph 2.4 and 2.5).
For the presentation of the RADseq results we start with a descriptive paragraph, providing additional information about the sensitivity to the diversity of parameters used to filter RADseq data by the bioinformatic pipeline, before presenting the results of the population genetics approaches. We did so, because filtering RADseq data and adjusting filtering parameters does not yet follow a standardized protocol and no consensus has been reached. As such we believe that it is important to present such aspects before dwelling into the biological results of the RADseq.
<<<
Fig. 1a caption “The volcanic active zone is marked with curved dotted lines”: as for me, these lines are quite hard to see. I only realized where they were after looking at the map in Kornobis et al. 2010. Maybe it’s worth changing the shade of gray (to white?), or the thickness, or the dotting patter to make them more visible.
>>> Our Reply 9: We agree and we changed the figure by increasing the grey contrast between the volcanic active zone and the other features on the map.
<<<
Fig. 1a: why is latitude marked in ° O (not W)?
>>> Our Reply 10: O stands for Occidental, but we agree to follow the more common approach using E for east. Changing from occidental to east, the longitudinal coordinates for Iceland become negatives.
<<<
L405 perfectly support => perfectly supportED (or supports)?
>>> Our Reply 11: We changed to “supports”. <<<
L416 overlapPED?
>>> Our Reply 12: We changed it. <<<
Reviewer 2 Report
This study should be published immediately as it is. It is very well written, easy to follow and to understand. The methods that were used are state-of-the-art and the topic on the evolution and adaptation of these groundwater amphipods is interesting and inciting. I congratulate the authors for this study.
Author Response
Reviewer 2
This study should be published immediately as it is. It is very well written, easy to follow and to understand. The methods that were used are state-of-the-art and the topic on the evolution and adaptation of these groundwater amphipods is interesting and inciting. I congratulate the authors for this study.
>>> Our Reply 13: We appreciate this very positive comment about our work. <<<
Reviewer 3 Report
This is a splendid study using navel genomic approach to phylogeography of Crangonyx islandicus, and deal with incongruent mitochondrial and nuclear phylogenies. Authors use cautious approach, are aware of used methods drawbacks and take steps to mitigate them.
Overall, the manuscript is well written and presents the study in great detail. However, some parts received far less author attention than others. ddRADseq procedures and protocols descriptions are documented with exemplary attention to details, but M&M part addressing mitochondrial sequences is too brief.
Have you followed Kornobis et al. with no modifications - same kits and ABI sequencer model? ABI PRISM 3100 Genetic Analyser is rather old, and not many are still in working condition. Have you followed same PCR protocols? (there is no mention of PCR at all, as if you sequenced genomic DNA directly with no amplification) This part needs to be either expanded and include more details, or explicitly say you done everything (not just the Sanger sequencing reaction) the same way.
Other minor issue is the first sentence of the abstract - it is a very awkward and confusing. Please rephrase.
Author Response
Reviewer 3 :
Overall, the manuscript is well written and presents the study in great detail. However, some parts received far less author attention than others. ddRADseq procedures and protocols descriptions are documented with exemplary attention to details, but M&M part addressing mitochondrial sequences is too brief.
Have you followed Kornobis et al. with no modifications - same kits and ABI sequencer model? ABI PRISM 3100 Genetic Analyser is rather old, and not many are still in working condition. Have you followed same PCR protocols? (there is no mention of PCR at all, as if you sequenced genomic DNA directly with no amplification) This part needs to be either expanded and include more details, or explicitly say you done everything (not just the Sanger sequencing reaction) the same way.
>>> Our Reply 14: We agree that the description was too short and confusing. Indeed we followed the exact same protocol for PCR amplification of the CO1 and 16S gene but the sequences were run on Genetic Analyser (3500xL Applied Biosystem).
We provided additional information to clarify the protocol L.169-172. “For PCR amplifications of the CO1 and 16S genes for the 21 new individuals, we followed the same protocol described in Kornobis et al. [8], and the sequencing was performed in both directions using the Sanger method on Genetic Analyser (3500xL Applied Biosystem).”
<<<
Other minor issue is the first sentence of the abstract - it is a very awkward and confusing. Please rephrase.
>>> Our Reply 15: We rephrased this sentence to improve the clarity L.19-21 “Analysis of phylogeographic patterns have often been based on mitochondrial DNA variation, but recent analyses including nuclear DNA have revealed mito-nuclear discordances and complex evolutionary histories.”
<<<
Reviewer 4 Report
Review
Paper title: Contrasting phylogeographic patterns of mitochondrial and genomic variation in the groundwater amphipod (Crangonyx islandicus) that survived the Ice age in Iceland
The authors sampled individuals of Crangonyx islandicus representing three distinct mitochondrial clades in Iceland to revisit the phylogeography of this endemic groundwater amphipod species and test the likelihood of the mitochondrial versus ITS scenario. The authors evaluated the mitochondrial scenario using a Bayesian structured coalescent analysis, tested the population structure using a number of indices, evaluated the level of incongruence among loci, and tested the amount of admixture between sampling locations caused by introgression or incomplete lineage sorting. In general, the authors’ results support the ITS scenario. The introgression led to male-biased dispersal among northern locations or mitochondrial capture. These results expand our knowledge of the life-history traits of the Crangonyx islandicus populations.
All these reasons explain the relevance of the paper by David Eme and co-authors submitted to "Diversity".
General scores.
The data presented by the authors are original and significant. The study is correctly designed and the authors used appropriate sampling methods. In general, statistical analyses are performed with good technical standards. The authors conducted careful work that may attract the attention of a wide range of specialists focused on crustacean phylogeography.
Recommendations.
Table S4.1 should be updated with sampling dates for each site.
Specific remarks.
L 13. Consider replacing “were commonly” with “was commonly”
L 35. Consider replacing “offer” with “offers”
L 36. Consider replacing “are the main goals” with “is the main goal”
L 42. Consider replacing “have singular” with “have a singular”
L 68. Consider replacing “sex biased” with “sex-biased”
L 108. Consider replacing “at their infancy” with “in their infancy”
L 115. Consider replacing “are found” with “is found”
L 131. Consider replacing “discrepancy of” with “discrepancy between”
L 152. Consider replacing “were stored in” with “were fixed with”
L 164. Consider replacing “phenol chloroform” with “phenol-chloroform”
L 174. Consider replacing “hours at” with “hours at the”
L 191. Consider replacing “sequencing following” with “sequencing following the”
L 249. Consider replacing “coverage dataset” with “coverage datasets”
L 285. Consider replacing “corrected for” with “corrected for a”
L 312. Consider replacing “the p-value of each loci were combined following the” with “the p-values of each loci were combined following”
L 317. Consider replacing “were evaluated” with “was evaluated”
L 318. Consider replacing “the p-value” with “the p-values”
L 319. Consider replacing “testing according” with “testing according to”
L 344. Consider replacing “lineage” with “lineages”
L 349. Consider replacing “It allows estimating” with “It allows for estimating”
L 359. Consider replacing “The first 10% of each chain was discarded and the remaining of both chains was” with “The first 10% of each chain were discarded and the remaining of both chains were”
L 368. Consider replacing “compare two models” with “compare the two models”
L 382. Consider replacing “two alternative scenario” with “two alternative scenarios
L 393. Consider replacing “For each of this combination” with “For each of these combinations”
L 405. Consider replacing “support” with “supports”
L 408. Consider replacing “north west locations” with “northwestern locations”
L 413. Consider replacing “north east population” with “northeastern population”
L 418. Consider replacing “have higher” with “have a higher”
L 431. Consider replacing “The difference of” with “The difference in”
L 485. Consider replacing “showed higher” with “showed a higher”
L 510. Consider replacing “in south-” with “in the south-”
L 534. Consider replacing “a missing data” with “missing data”
L 535. Consider replacing “plots” with “plots in”
L 536. Consider replacing “plots” with “plots in”
L 539. Consider replacing “are displayed” with “is displayed”
L 540. Consider replacing “the mains” with “the main”
L 555. Consider replacing “overlaps” with “overlap”
L 577. Consider replacing “north east” with “northeastern”
L 600. Consider replacing “the dataset 3” with “datasets 3”
L 625. Consider replacing “support by” with “support from”
L 637. Consider replacing “Our results didn’t reveal a systematic bias for a particular differentiation metrics” with “Our results reveal no systematic bias for particular differentiation metrics”
L 640. Consider replacing “increase” with “increases”
L 652. Consider replacing “to assess” with “to assessing”
L 658. Consider replacing “estimate” with “estimating”
L 661. Consider replacing “agreement to” with “agreement with”
L 665. Consider replacing “the read with a low coverage are” with “a read with a low coverage is”
L 674. Consider replacing “rate” with “rates”
L 677. Consider replacing “larger genome size than their surface sister species due to reduction” with “larger genome sizes than their surface sister species due to the reduction”
L 693. Consider replacing “are becoming” with “is becoming”
L 719. Consider replacing “lower nuclear substitution rate” with “a lower nuclear substitution rate”
L 721. Consider replacing “mitochondrial genome” with “the mitochondrial genome”
L 722. Consider replacing “nuclear genome” with “nuclear genomes”
L 729. Consider replacing “much quickly” with “much more quickly”
L 746. Consider replacing “north and northeast” with “northern and northeastern”
L 749. Consider replacing “male biased” with “male-biased”
L 752. Consider replacing “largest variation was observed at highest” with “the largest variation was observed at higher”
L 757. Consider replacing “of last glacial” with “of the last glacial”
L 766. Consider replacing “Similar scenario” with “A similar scenario”
L 785. Consider replacing “large population” with “a large population”
L 794. Consider replacing “divergence on” with “divergence in”
L 795. Consider replacing “north” with “northern”
L 797. Consider replacing “or being captured” with “or be captured”
L 810. Consider replacing “information of” with “information on”
Author Response
Reviewer 4 :
Review
Paper title: Contrasting phylogeographic patterns of mitochondrial and genomic variation in the groundwater amphipod (Crangonyx islandicus) that survived the Ice age in Iceland
The authors sampled individuals of Crangonyx islandicus representing three distinct mitochondrial clades in Iceland to revisit the phylogeography of this endemic groundwater amphipod species and test the likelihood of the mitochondrial versus ITS scenario. The authors evaluated the mitochondrial scenario using a Bayesian structured coalescent analysis, tested the population structure using a number of indices, evaluated the level of incongruence among loci, and tested the amount of admixture between sampling locations caused by introgression or incomplete lineage sorting. In general, the authors’ results support the ITS scenario. The introgression led to male-biased dispersal among northern locations or mitochondrial capture. These results expand our knowledge of the life-history traits of the Crangonyx islandicus populations.
All these reasons explain the relevance of the paper by David Eme and co-authors submitted to "Diversity".
General scores.
The data presented by the authors are original and significant. The study is correctly designed and the authors used appropriate sampling methods. In general, statistical analyses are performed with good technical standards. The authors conducted careful work that may attract the attention of a wide range of specialists focused on crustacean phylogeography.
>>> Our Reply 16: We appreciate the positive appraisal of our work. <<<
Recommendations.
Table S4.1 should be updated with sampling dates for each site.
>>> Our Reply 17: We provided the sampling date of the sampling site in Table S4.1 as well as in Table S1.1 and in the new Table S1.2.
<<<
Specific remarks.
L 13. Consider replacing “were commonly” with “was commonly”
>>> We changed the sentence. <<<
L 35. Consider replacing “offer” with “offers”
>>> Done. <<<
L 36. Consider replacing “are the main goals” with “is the main goal”
>>> Done. <<<
L 42. Consider replacing “have singular” with “have a singular”
>>> Done. <<<
L 68. Consider replacing “sex biased” with “sex-biased”
>>> Done. <<<
L 108. Consider replacing “at their infancy” with “in their infancy”
>>> Done. <<<
L 115. Consider replacing “are found” with “is found”
>>> We kept “are” as there are two species. <<<
L 131. Consider replacing “discrepancy of” with “discrepancy between”
>>> Done. <<<
L 152. Consider replacing “were stored in” with “were fixed with”
>>> Done. <<<
L 164. Consider replacing “phenol chloroform” with “phenol-chloroform”
>>> Done. <<<
L 174. Consider replacing “hours at” with “hours at the”
>>> Done. <<<
L 191. Consider replacing “sequencing following” with “sequencing following the”
>>> Done. <<<
L 249. Consider replacing “coverage dataset” with “coverage datasets”
>>> Done. <<<
L 285. Consider replacing “corrected for” with “corrected for a”
>>> Done. <<<
L 312. Consider replacing “the p-value of each loci were combined following the” with “the p-values of each loci were combined following”
>>> Done. <<<
L 317. Consider replacing “were evaluated” with “was evaluated”
>>> Done. <<<
L 318. Consider replacing “the p-value” with “the p-values”
>>> Done. <<<
L 319. Consider replacing “testing according” with “testing according to”
>>> Done. <<<
L 344. Consider replacing “lineage” with “lineages”
>>> Done. <<<
L 349. Consider replacing “It allows estimating” with “It allows for estimating”
>>> Done. <<<
L 359. Consider replacing “The first 10% of each chain was discarded and the remaining of both chains was” with “The first 10% of each chain were discarded and the remaining of both chains were”
>>> Done. <<<
L 368. Consider replacing “compare two models” with “compare the two models”
>>> Done. <<<
L 382. Consider replacing “two alternative scenario” with “two alternative scenarios
>>> Done. <<<
L 393. Consider replacing “For each of this combination” with “For each of these combinations”
>>> Done. <<<
L 405. Consider replacing “support” with “supports”
>>> Done. <<<
L 408. Consider replacing “north west locations” with “northwestern locations”
>>> We changed to “northern locations”. <<<
L 413. Consider replacing “north east population” with “northeastern population”
>>> Done. <<<
L 418. Consider replacing “have higher” with “have a higher”
>>> Done. <<<
L 431. Consider replacing “The difference of” with “The difference in”
>>> Done. <<<
L 485. Consider replacing “showed higher” with “showed a higher”
>>> Done. <<<
L 510. Consider replacing “in south-” with “in the south-”
>>> Done. <<<
L 534. Consider replacing “a missing data” with “missing data”
>>> Done. <<<
L 535. Consider replacing “plots” with “plots in”
>>> Done. <<<
L 536. Consider replacing “plots” with “plots in”
>>> Done. <<<
L 539. Consider replacing “are displayed” with “is displayed”
>>> Done. <<<
L 540. Consider replacing “the mains” with “the main”
>>> Done. <<<
L 555. Consider replacing “overlaps” with “overlap”
>>> Done. <<<
L 577. Consider replacing “north east” with “northeastern”
>>> Done. <<<
L 600. Consider replacing “the dataset 3” with “datasets 3”
>>> Done. <<<
L 625. Consider replacing “support by” with “support from”
>>> Done. <<<
L 637. Consider replacing “Our results didn’t reveal a systematic bias for a particular differentiation metrics” with “Our results reveal no systematic bias for particular differentiation metrics”
>>> Done. <<<
L 640. Consider replacing “increase” with “increases”
>>> Done. <<<
L 652. Consider replacing “to assess” with “to assessing”
>>> Done. <<<
L 658. Consider replacing “estimate” with “estimating”
>>> Done. <<<
L 661. Consider replacing “agreement to” with “agreement with”
>>> Done. <<<
L 665. Consider replacing “the read with a low coverage are” with “a read with a low coverage is”
>>> We change by “the reads with a low coverage are”. <<<
L 674. Consider replacing “rate” with “rates”
>>> Done. <<<
L 677. Consider replacing “larger genome size than their surface sister species due to reduction” with “larger genome sizes than their surface sister species due to the reduction”
>>> Done. <<<
L 693. Consider replacing “are becoming” with “is becoming”
>>> Done. <<<
L 719. Consider replacing “lower nuclear substitution rate” with “a lower nuclear substitution rate”
>>> Done. <<<
L 721. Consider replacing “mitochondrial genome” with “the mitochondrial genome”
>>> Done. <<<
L 722. Consider replacing “nuclear genome” with “nuclear genomes”
>>> Done. <<<
L 729. Consider replacing “much quickly” with “much more quickly”
>>> Done. <<<
L 746. Consider replacing “north and northeast” with “northern and northeastern”
>>> Done. <<<
L 749. Consider replacing “male biased” with “male-biased”
>>> Done. <<<
L 752. Consider replacing “largest variation was observed at highest” with “the largest variation was observed at higher”
>>> Done. <<<
L 757. Consider replacing “of last glacial” with “of the last glacial”
>>> Done. <<<
L 766. Consider replacing “Similar scenario” with “A similar scenario”
>>> Done. <<<
L 785. Consider replacing “large population” with “a large population”
>>> Done. <<<
L 794. Consider replacing “divergence on” with “divergence in”
>>> Done. <<<
L 795. Consider replacing “north” with “northern”
>>> Done. <<<
L 797. Consider replacing “or being captured” with “or be captured”
>>> Done. <<<
L 810. Consider replacing “information of” with “information on”
>>> Done. <<<
Round 2
Reviewer 1 Report
The authors did a great job improving the manuscript and reproducibility of the data analysis. I'm totally satisfied with the revision and looking forward to seeing the paper published.
Author Response
Thank you.
A revised version has been submitted